# Potential of Transcript Editing Across Mitogenomes of Early Land Plants Shows Novel and Familiar Trends

**DOI:** 10.3390/ijms20122963

**Published:** 2019-06-18

**Authors:** Kamil Myszczyński, Monika Ślipiko, Jakub Sawicki

**Affiliations:** Department of Botany and Nature Protection, University of Warmia and Mazury in Olsztyn, Plac Łódzki 1, 10-727 Olsztyn, Poland; monika.slipiko@uwm.edu.pl (M.Ś.); jakub.sawicki@uwm.edu.pl (J.S.)

**Keywords:** RNA editing, early land plants, mitochondrial genome, liverwort, moss

## Abstract

RNA editing alters the identity of nucleotides in an RNA sequence so that the mature transcript differs from the template defined in the genome. This process has been observed in chloroplasts and mitochondria of both seed and early land plants. However, the frequency of RNA editing in plant mitochondria ranges from zero to thousands of editing sites. To date, analyses of RNA editing in mitochondria of early land plants have been conducted on a small number of genes or mitochondrial genomes of a single species. This study provides an overview of the mitogenomic RNA editing potential of the main lineages of these two groups of early land plants by predicting the RNA editing sites of 33 mitochondrial genes of 37 species of liverworts and mosses. For the purpose of the research, we newly assembled seven mitochondrial genomes of liverworts. The total number of liverwort genera with known complete mitogenome sequences has doubled and, as a result, the available complete mitogenome sequences now span almost all orders of liverworts. The RNA editing site predictions revealed that C-to-U RNA editing in liverworts and mosses is group-specific. This is especially evident in the case of liverwort lineages. The average level of C-to-U RNA editing appears to be over three times higher in liverworts than in mosses, while the C-to-U editing frequency of the majority of genes seems to be consistent for each gene across bryophytes.

## 1. Introduction

RNA editing is a modification of transcripts encoded by organellar and nuclear genomes that occurs in various organisms, including animals, plants, fungi, and protists [1,2,3,4]. The modifications of transcripts caused by the RNA editing effect with encoding of alternative amino acid sequences is necessary for correct functioning of some protein-coding genes [5]. The RNA editing may be also involved in increasing genetic diversity and adaptation [6]. In plants, this process appears mainly in the protein-coding regions of a genome, but several cases of RNA editing of structural RNAs, introns, or spacers have also been reported [7,8]. The editing process may generate single amino acid differences and can also influence the gene expression by the generation of new start and stop codons [9,10] and could be required for successful intron splicing process [11,12]. On the other hand, RNA editing also occurs at sites where the editing substitution is not essential for protein expression or activity. However, the occurrence of RNA editing substitutions might vary across living conditions i.e., stage of development, environmental conditions [13]. Furthermore, the editing substitutions were identified at intergenic spacers [14]. In plants, RNA editing seems to be more common in mitogenomes than plastomes [9,15], although the frequency and type of editing is assumed as species specific [16].

The RNA editing process is known in most currently living plant lineages [17,18,19], including the early land plants—bryophytes [16,20,21,22]. Bryophytes comprise three lineages, hornworts, liverworts, and mosses that share a common haploid-dominant life cycle. The monophyly of bryophytes and their evolutionary relationships with tracheophytes were the subject of several phylogenomic studies, resulting in competing and supported hypotheses [23,24,25,26]. However, all most likely scenarios resolved liverworts as a sister group to mosses and differ in the evolutionary position of hornworts. The backbone phylogeny of mosses resolved using phylogenomic data from three genomes identified all major evolutionary units within this group [27]. The liverwort phylogeny was not widely analyzed using a phylogenomics approach, but limited mitogenomic data [20,28] is congruent with earlier phylogenetics studies based on Sanger-derived (and thus limited) nucleotide datasets [29], which resolves four major clades, including early-diverging Haplomitriopsida, complex thalloids (Marchantiopsida), simple thalloids, and leafy liverworts.

The frequency of RNA editing is poorly explored in the context of plant evolutionary diversification. In the plastomes of angiosperms, the RNA editing seems to decrease during diversification [17] and several factors may be behind this process. One of the best-documented is the correlation of the rate of RNA editing with the diversification of DYW-Type pentatricopeptide (PPR) genes playing a main role in this process, although the available data is limited to only a few species [16,30,31,32].

Due to limited data availability, the evolutionary patterns of RNA editing in the mitogenomes of early land plants are poorly explored. The analysis of three mitochondrial *nad* genes (*nad*2, *nad*4, and *nad*5) revealed a decrease in RNA editing frequency during diversification of liverworts and mosses [16]. In the angiosperms and ferns, these posttranscriptional modifications appear mainly in mitochondrial transcripts, while in the plastomes this phenomenon seems to be limited [18,19]. More frequent RNA editing in mitogenomes than in plastomes was also confirmed in mosses [21,22], although in the sister lineage, liverworts, the plastomes seems to be more abundant in editing sites than mitogenomes [20].

The backbone mitophylogeny of mosses and liverworts allows for phylogenetic insight into the evolution of quantity and type (C-to-U or U-to-C) of mitochondrial RNA editing. Earlier studies revealed mixed patterns of this phenomenon in bryophytes, but the datasets were usually very limited in terms of studied genes [16].

The standard procedure used in the discovery of RNA editing sites is direct comparison of the RNA transcript sequence, usually obtained via RT-PCR or RNA-Seq methods, with genomic DNA sequence [33]. However, extensive analyses of RNA editing in hundreds of genes may be very costly and time-consuming. An alternative to experimental analyses is the use of bioinformatic methods. RNA editing sites in translated regions can be predicted by a comparison of amino acid sequences deduced from genomic DNA sequences [6]. Such predictions are based on comparison of a gene sequence to its respective homologue in non-editing or low-editing taxa [34]. The accuracy of RNA editing site predictions depends largely on the accessibility of closely related species used as references which preferably show little or no evidence of RNA editing. The analyses conducted by Rüdinger et al. (2012) [16] have shown that in the majority of studied cases, the use of the closest related species as a reference gives the best results (in terms of RNA editing event predictions) when compared with experimentally confirmed RNA editing sites. To date, RNA editing sites can be predicted in plant mitochondria and chloroplasts by bioinformatic tools such as PREPACT 3.12.0 [35].

In this study, the predicted RNA editing patterns of 33 mitochondrial genes from 36 bryophytes were compared, comprising 18 liverworts and 18 mosses (Appendix A), representing all major phylogenetic groups. Since the number of known bryophytes mitogenomes was strongly biased towards mosses, for the purpose of this study seven liverworts mitogenomes were newly sequenced, fully assembled, and annotated. The mitogenomes sequenced for this study were obtained for all major evolutionary lineages, including complex thalloids, simple thalloids, and leafy liverworts. Additionally, for two species of liverworts, where the complete mitogenome sequences could not be assembled, the sets of 33 protein-coding sequences of mitochondrial genes were identified and integrated in the study. The genome-wide study helped to answer several questions: are differences in the number of editing sites among the main lineages of liverworts and mosses significant? Does the pattern of RNA editing site gain/loss reveal any evolutionary pattern? Is RNA editing biased towards specific genes?

## 2. Results and Discussion

### 2.1. Characteristics of Newly Sequenced Liverwort Mitogenomes

For the purpose of this study, seven mitochondrial genomes of liverworts were newly sequenced, assembled, and annotated: *Fossombronia foveolata*, *Haplomitrium hookeri*, *Jungermannia sphaerocarpa*, *Metzgeria furcata*, *Porella platyphylla*, *Ptilidium ciliare*, and *Riccia fluitans* (Appendix A).

The mitogenome size of the seven species ranged from 156,050 bp (*J. sphaerocarpa*) to 185,621 bp (*R. fluitans*), and the overall GC content varied from 44.8% (*J. sphaerocarpa* and *P. ciliare*) to 47.7% (*H. hookeri*) (Table 1). Both mitochondrial genome sizes and GC contents are consistent with previous studies on liverworts [20,36,37,38,39,40].

The mitochondrial genomes of the seven species encoded 72–74 genes, including an identical set of 42 protein-coding genes, 27–28 tRNA genes, 3 rRNA genes, and 1 pseudogene (*nad7*). However, as in the previously studied species of Halpomitriopsida clade (*Treubia lacunosa* and *Haplomitrium mnioides*) [30], the *nad7* gene was also identified as functional, containing three exons, in *H. hookeri*. Other differences in gene content are due to lack of a *nad7* pseudogene (*J. sphaerocarpa*), *trnR-UCG* gene (*H. hookeri*), and a second copy of the *trnMf-CAU* gene (*F. foveolata* and *J. sphaerocarpa*).

The number of genes containing introns ranges from 14 to 16 and the total number of introns within the mitogenomes varied from 25 (*F. foveolata*) to 30 (*H. hookeri* and *R. fluitans*). Although the mitogenomes of the seven liverworts species contain over a dozen of genes consisting of introns, the discrepancy in the overall intron number between species results from just four genes: *atp1*, *cox1, nad7*, and *rrn18*. The gene order of the seven species’ mitogenomes is similar to the other aforementioned species of liverworts [36,37,38,39,40].

Additionally, for *Pallavicinia lyellii* and *Pellia epiphylla*, where the complete mitogenome sequences could not be assembled, the sets of 33 protein-coding sequences of mitochondrial genes were identified and integrated in the study. Assembled CDS include such genes as: *atp4*, *atp6*, *atp8*, *atp9*, *cox1*, *cox2*, *cox3*, *nad1*, *nad2*, *nad3*, *nad4*, *nad4L*, *nad5*, *nad6*, *nad9*, *rpl2*, *rpl5*, *rpl6*, *rpl10*, *rpl16*, *rps1*, *rps2*, *rps3*, *rps4*, *rps7*, *rps11*, *rps12*, *rps13*, *rps14*, *rps19*, *sdh3*, *sdh4*, and *tatC*. The analyses were limited to only protein-coding genes commonly occurring in both liverwort and moss mitochondria. 

Seven mitochondrial genomes obtained in this study double the total number of liverwort genera with known complete mitogenome sequences. Moreover, two sets of mitochondrial protein-coding sequences (*Pallavicinia lyellii* and *Pellia epiphylla*) and two complete mitogenomic sequences (*Fossombronia foveolata* and *Metzgeria furcata*) of simple thalloid provide a great extent of the sequence data of this poorly explored (in terms of mitochondrial genomes) group of liverworts. To date, the only representative of simple thalloid liverworts with a completely known mitogenome sequence was *Aneura pinguis* [39]. All of these newly-sequenced mitogenomes of liverworts, together with ones that were previously known, spanning through almost all orders of liverworts (excluding Neohodgsoniales and Sphaerocarpales), provide an extensive source of knowledge of Marchantiophyta mitochondrial genomics.

### 2.2. RNA Editing Predictions in Liverwort Mitochondria

For several genes, a significant drop in predicted editing sites appears in specific taxonomic groups. Besides the known, non-editing model liverwort genus *Marchantia* [31], two other complex thalloid liverworts, *Blasia pusilla* and *Riccia fluitans*, were found to have the lowest editing potential among liverworts. Only 18 RNA editing sites were predicted in the *R. fluitans* mitogenome (11 C-to-U and 7 U-to-C) and 30 were found in *B. pusilla* (17 C-to-U and 13 U-to-C). In both mitogenomes, the share of reverse U-to-C editing sites in total number was high, unlike in other known mitogenomes of liverworts and angiosperms [41,42]. However, the total number of predicted RNA editing sites within those two mitogenomes was low in comparison to the rest of the investigated liverworts.

The total number of predicted RNA editing sites declined in the leafy liverworts, which are a late-diverging group (Figure 1).

The trend of decreasing number of predicted editing sites along with diversification observed in the leafy liverworts group could be connected with structural changes of the mitochondrial genes, i.e., intron losses [20,40]. The comparative analysis of *Calypogeia* and *Tritomaria* mitochondrial genes with other known liverwort mitogenomes revealed precise intron loss [40], which is characteristic for losing introns by retroprocessing [43,44]. It is not clear if retroprocessing triggers editing site losses [19], but some cases of losing editing sites adjacent to lost introns have been previously reported [44,45,46,47,48]. Since one-third of liverwort protein-coding genes contain introns (14 out of 42 CDS) the retroprocessing, as a main mechanism of losing introns, can play a significant role in decreasing the number of editing sites. This hypothesis could be re-evaluated when more mitogenomes from leafy liverworts become available to shed more light on the intron loss pattern and its correlation with a reduction of editing sites. The loss of editing sites comprises only C-to-U events, the number of predicted U-to-C editing sites remains stable across all liverwort lineages (Figure 1). This may suggest different functional relevance of both types of editing or differences in the accuracy of predictions between C-to-U and U-to-C editing.

However, not all species and genes fit into the above scenario. A low number of editing sites (89) were detected in the early divergent liverwort, *Treubia lacunosa*, while a sister to this species, *Haplomitrium hookeri*, was the richest in the editing site species with a total of 732, including 712 C-to-U and 20 U-to-C sites. This was also confirmed in the previous studies on *nad4* and *nad5* genes [16], where 55 and 54 editing sites were identified, respectively, in *Haplomitrium mnioides,* suggesting a comparable level of RNA editing as predicted in *H. hookeri*.

The C-to-U editing frequency, expressed as the number of predicted C-to-U editing sites per gene CDS length, was calculated for each of the 33 protein-coding genes of 17 species of liverworts (see method section for details). The U-to-C editing frequency was calculated analogously. For every liverwort group (complex, leafy, and simple thalloid), the average editing frequencies were calculated and defined as mean C-to-U or U-to-C editing frequencies of each gene (Appendix A). However, further analyses were mainly focused on leafy and simple thalloid liverworts since, as expected, complex liverwort genes present hardly any predicted RNA editing. 

The observed mean C-to-U editing frequency was not consistent across genes, ranging from 0% in *rps13* to 2.17% in *atp9* and from 0.32% in *rps12* to 4.98% in *atp9* in leafy and simple thalloid liverworts, respectively (Figure 2). Fourteen out of 33 analyzed mitochondrial genes did not reveal significant (Kruskal–Wallis test, *p*-value < 0.05) differences between the means of C-to-U editing frequency in mitogenomes of simple thalloids and leafy liverworts (Appendix A), although in all cases more editing sites were predicted in simple thalloid than in leafy liverworts (Figure 2).

For nine of them (*atp*4, *nad*9, *rpl*2, *rpl*5, *rps1*, *rps3*, *rps7*, *rps12*, and *rps14*), the number of predicted editing sites was relatively low and stable among analyzed mitogenomes of leafy and simple thalloid liverworts. The most notable decrease (3-fold or more) in editing frequency in leafy compared to simple thalloid liverworts was found in *atp6*, *cox3*, *nad4*, *sdh4*, *cox1, rpl6*, and *rps4* genes (Appendix A). However, the differences in C-to-U mean editing frequencies are mainly due to overall level differences in C-to-U predicted RNA editing between leafy and simple thalloid liverworts. The average number of C-to-U predicted editing sites was 2.6 times lower in leafy than simple thalloid liverworts. The majority of genes that presented a high C-to-U editing frequency in leafy liverworts also presented high C-to-U editing frequency in simple thalloid liverworts and, consequently, genes that presented a low C-to-U editing frequency in one group also presented a low C-to-U editing frequency in the other group.

A great majority of the analyzed mitochondrial genes (31 out of 33) did not reveal significant (Kruskal–Wallis test, *p*-value < 0.05) differences between means of U-to-C editing frequency in the mitogenomes of simple and leafy liverworts (Appendix A). Moreover, contrary to C-to-U editing frequencies, the U-to-C editing frequencies were similarly distributed across the genes of both groups of liverworts. Overall, the levels of U-to-C RNA editing were much lower than C-to-U across all liverwort lineages (except complex thalloids and *T. lacunosa*, in which the total numbers of predicted editing sites were generally low) (Figure 1 and Appendix A). However, the homogeneous distribution of U-to-C editing frequencies are likely to be overestimates due to lower accuracy of the software for U-to-C sites predictions.

### 2.3. RNA Editing Predictions in Moss Mitochondria

Although the evolutionary sister to the liverworts group, mosses, are not included on the list of editing sites during the diversification scenario, the general rule of higher frequency of editing sites in the early-diverging lineages can be assumed. The early mosses group displayed the highest overall number of predicted editing sites, as well as C-to-U and U-to-C substitutions calculated separately (Figure 1). The average percentage of predicted editing sites was 1.85 times higher in plagiotropic mosses than earlier-diverging, orthotropic mosses. The difference is mainly due to predicted C-to-U substitutions (2.3 times higher in plagiotropic mosses) while the average percentage of predicted U-to-C substitutions was similar in both groups. The mitogenome of *Ptychomnion cygnisetum*, the representative of plagiotropic mosses, also contained the most C-to-U predicted editing sites (124) of all studied moss species. In contrast, *Funaria hygrometrica* mitogenome, with just 9 C-to-U and 5 U-to-C predicted editing sites, was the least editing-rich species (Figure 1 and Appendix A). Consistently, *Physcomitrella patens*, another member of Funariaceae family, has been previously proven to also contain low rates of RNA editing [21,49].

The C-to-U editing frequency and U-to-C editing were predicted for each of the 33 protein-coding genes of 18 species of mosses (see method section for details). For every moss group (early-diverging, orthotropic, and plagiotropic), the average editing frequencies were calculated and defined as mean C-to-U or U-to-C editing frequencies of each gene (Appendix A). 

The means of C-to-U editing frequency of genes were the highest for 18 genes of early mosses, 14 genes of plagiotropic mosses and one gene of orthotropic mosses. Only 8 out of 33 protein-coding genes’ editing frequencies significantly differed among the early and plagiotropic mosses (Appendix A). Moss C-to-U editing frequencies, as in the case of liverworts, were not consistent across genes. The mean C-to-U editing frequencies ranged from 0% in *rpl16* and *nad4L* to 1.32% in *tatC*, from 0% in *atp4*, *nad2*, *nad4L*, *rpl16*, *rps12*, *rps13*, and *rps19* to 0.73% in *atp9* and from 0% in *rps2* to 0.96% in *atp9* in early-diverging, orthotropic, and plagiotropic mosses, respectively (Figure 3).

The C-to-U editing frequency differed significantly (Kruskal–Wallis test, *p*-value < 0.05) between at least two groups of mosses in 20 out of 33 protein-coding genes. Hence, the mosses seem to be a more homogenic group of plants, in terms of RNA editing, than liverworts, in which 31 protein-coding genes varied significantly (Kruskal–Wallis test, *p*-value < 0.05) between at least two groups in predicted C-to-U substitutions (Appendix A). Moreover, 14 protein-coding genes did not reveal significant differences between editing frequencies in the mitogenomes of any two groups of mosses while, for liverworts, only two protein-coding genes did not reveal significant differences between editing frequencies in the mitogenomes of any two groups. Considering the 20 genes that presented a different number of predicted C-to-U editing sites between at least two groups of mosses, 13 genes were identified among early and orthotropic mosses, 11 genes among orthotropic and plagiotropic mosses, and 8 genes among early and plagiotropic mosses. The most notable decrease (3-fold or more) in editing frequency in plagiotropic compared to orthotropic mosses was found in *tatC*, *rpl2*, *nad1, nad3*, *rps7*, *nad5*, and *rpl10* genes (Figure 3). The differences in C-to-U mean editing frequencies, although not as explicit as in liverwort groups, were mainly due to the overall level differences in C-to-U predicted RNA editing between orthotropic and plagiotropic mosses. However, the C-to-U editing frequencies of each gene against the overall level of C-to-U editing seem to be consistent across species of both orthotropic and plagiotropic mosses.

The differences between overall levels of C-to-U and U-to-C predicted RNA editing were not as clear as in the case of liverwort mitogenomes. In fact, the levels of the two types of RNA editing were nearly equal across all lineages of mosses. 

Despite this, the U-to-C RNA editing predictions [35] should be treated with caution as the verification of RNA editing sites with four database-derived transcriptomes of liverworts and mosses [50] indicated that there are hardly any confirmed U-to-C predictions, i.e., the ratios of predicted editing sites to those confirmed via database-derived transcriptome were: 0 to 12, 0 to 34, 3 to 6, and 0 to 5 in *A. angustatum*, *B. aphylla*, *P. lyellii*, and *T. pellucida* respectively. While in the case of C-to-U RNA editing sites, the ratios were as follows: 10 to 20, 9 to 49, 225 to 250, and 2 to 7 in *A. angustatum*, *B. aphylla*, *P. lyellii*, and *T. pellucida*, respectively (Appendix A). Hence, further analyses of RNA editing in liverworts and mosses were restricted mainly to C-to-U editing sites and it should be noted that some discrepancy between the obtained dataset and database-derived transcriptomes is to be expected, not only due to the use of different individuals for RNA editing site predictions and transcriptome analyses. Recent studies have shown that RNA editing patterns and level differences may be caused by stress factors [51,52,53,54,55], diseases [56,57], as well as different composition of cell populations [58] and tissues [59]. Considering the aforementioned issues, some similarity in RNA editing site distribution between genome-based predictions and database-derived transcripts was expected. Nevertheless, to unambiguously confirm RNA editing events of particular species, one should examine protein-coding genes and corresponding transcriptomes derived at a specific time point from the same sample, i.e., a specific tissue. Therefore, the aforementioned quasi-validation should be considered as additional information rather than conclusive evidence of RNA editing.

### 2.4. RNA Editing Across Mitochondrial Genes of Bryophytes

Since the species investigated in this study represent major phylogenetic groups of mosses and liverworts, this study analyzed whether mitochondrial protein-coding genes show consistent editing frequency across phylogeny. To explore the possibility that RNA editing potential of mosses and liverworts is biased towards specific genes or gene families, the mean C-to-U editing frequency of mitochondrial genes was investigated. The overall level of predicted C-to-U RNA editing was 3.5 times lower on average in mosses than in liverworts, while the overall level of predicted U-to-C RNA editing was 2.5 times higher on average in mosses than in liverworts. The correlation between C-to-U editing frequency of 33 mitochondrial genes of 18 liverworts and 18 mosses was moderately positive (*r* = 0.619, Pearson correlation, *p*-value = 0.0001211) across genes. A visual inspection of scaled means (liverwort to-moss scale factor = 1:3.57) of gene C-to-U editing frequencies (Figure 4) indicated that the correlation measure could be distorted by a particular group of genes.

The hypothesis that mean editing frequency is consistent for each gene across the two groups of early land plants seems to be true for the majority of mitochondrial genes studied in this research. This is well illustrated by the *atp9* gene, which showed high mean C-to-U editing frequency in both moss and liverworts relative to the overall predicted C-to-U editing level. Another example is the *rps12* gene where the mean C-to-U editing frequency is low among the two groups of early land plants (Figure 4). Previous studies focused on the RNA editing potential of genes of the *nad* family in liverwort and moss mitochondria revealed that the C-to-U RNA editing is taxon-dependent, rather than locus-dependent [16]. Up to now, research by Rüdinger et al. (2012) [16] was the only study that focused on such a broad spectrum of bryophyte species in terms of RNA editing events in mitochondrial genomes. However, the study investigated the editing patterns of just three genes: *nad5*, *nad4*, and *nad2* [16]. 

The current study revealed the moderate overall positive correlation (*r* = 0.619, Pearson correlation, *p*-value = 0.0001211) between a C-to-U editing frequency of 33 genes of liverwort and moss mitochondria. This suggests that, regardless the overall RNA editing level, the frequency of C-to-U RNA editing is preserved across genes of bryophytes. Although the majority of mitochondrial genes investigated in this study seem to be in accordance with this hypothesis, for some genes, like *nad2*, *nad4L*, *nad6*, *rps7*, and *rps14*, RNA editing seems to be taxon-dependent. Research focusing on the mitochondrial frequency of editing sites across phylogeny has not yet provided an unambiguous answer. A recent paper on 17 diverse angiosperms [19] has shown that the frequency of C-to-U editing does not seem to be fairly consistent for each gene across a species, although previous studies stand in contrast to this view [60]. Since there are substantial differences in the frequency of mitochondrial RNA editing among major plant lineages [18,47,48,60], one cannot assume in advance that the RNA editing patterns identified in one taxa apply to another. The results obtained in the current study show that *atp9* gene displays a high potential of C-to-U RNA editing frequency in bryophytes, whereas no editing sites were identified in the *atp9* of angiosperms [19,61,62].

The current observations on predicted RNA editing sites across mitogenomes of the two groups of early land plants suggest that the overall level of C-to-U RNA editing is over three times higher in liverworts than in mosses. Additionally, while investigating C-to-U RNA editing events in mitogenomes of early land plants, one can assume that mitochondrial genes such as *atp9*, *sdh3*, and *tatC* will show relatively high editing frequencies and, contrarily, *atp8*, *rpl16*, and *rps12* will present relatively low editing frequencies. Taken together, regardless of the species, some mitochondrial genes of early land plants display higher transcript editing potential than others.

### 2.5. Phylogenetic Relationships Based on Protein-Coding Mitochondrial Genes

The alignment of the set of 33 protein-coding sequences of mitochondrial genes common to liverworts and mosses (37 species in total) (Appendix A) was used to construct the phylogenetic tree. The phylogenetic analysis inferred from the partitioned mitochondrial CDS dataset clearly distinguished liverworts and mosses (Figure 1A) and is congruent with previous studies [28,63,64] based on multi-locus combined datasets. The mitogenomic analysis confirmed paraphylly on simple thalloid liverworts and resolved enigmatic leafy genus *Pleurozia* as a sister to Metzgeriales. For mosses, the RNA editing site distribution dataset analysis resulted in trees congruent with a CDS-based dataset, but with a lower resolving power and the presence of polytomies (Figure 1B). Phylogenetic analysis of RNA editing site distribution patterns of the liverworts resulted in trees partially incongruent with mitophylogenomic analysis (Figure 1B), mainly due to the position of complex thalloids/*Treubia* clade which resolved as a part of a leafy liverwort clade. Since in the mitogenomes of *Treubia*, *Blassia*, and *Riccia*, the numbers of RNA editing sites are very low (45, 17, and 11 C-to-U editing sites, respectively), the character-state matrix analysis could not be performed as well as for editing-site rich taxa. The second incongruence is the position of leafy *Pleurozia purpurea*. This taxon in the earlier studies was placed as a sister to Metzgeriales [65,66] and its position was also confirmed in mitophylogenomic analysis in this study (Figure 1B). The phylogenetic analysis based on RNA editing sites distribution pattern for the first time resolved *Pleurozia purpurea* within the leafy liverworts clade with good bootstrap values (84% maximum parsimony and 100% neighbor-joining bootstrap support) and provided result congruent with morphology. However, to provide better support for the leafy liverworts monophylly further investigation with the use of plastid and nuclear genes is required.

## 3. Materials and Methods 

### 3.1. Material

Total genomic DNA was extracted using ZR Plant/Seed DNA MiniPrepTM kit (Zymo Research Corp., Irvine, CA, USA) according to the manufacturer’s protocol from fresh tissue or herbarium specimens. The extracted DNA was quantified using a Qubit BR DNA kit (Thermo Fisher Scientific, Waltham, MA, USA). The genomic libraries were prepared using a TruSeq Nano DNA library preparation kit (Illumina, San Diego, CA, USA) and sequenced using HiSeqX sequencer (Illumina) by Macrogen Inc. (Seoul, Korea) to generate 150 bp paired-end reads with a 350 bp insert size between paired-ends. The list of GenBank accession numbers and newly sequenced species are given in Appendix A.

### 3.2. Mitochondrial Genome Assembly

Sequencing reads were cleaned by removing the adaptor sequences and low-quality reads with Trimmomatic 0.36 (The Usadel Lab, Aachen, Germany) [67]. The filtered reads were assembled using NOVOPlasty 2.7.2 (Interuniversity Institute of Bioinformatics in Brussels, Brussels, Belgium) [68] and Geneious 8.1 software (Biomatters Ltd., Auckland, New Zealand) [69].

The newly sequenced mitogenomes were annotated as described in the authors’ previous studies on liverworts [20,39,40]. In the case of two species, *Pellia epiphylla* and *Pallavicinia lyellii*, only complete CDS datasets were obtained, since the reliable assembly of complete mitogenomes was difficult due to the presence of tandem repeat-rich regions.

### 3.3. Prediction and Verification of RNA Editing Sites

The set of 33 protein-coding sequences of mitochondrial genes common to liverworts and mosses was extracted from newly-assembled mitogenomes as well as mitogenomes obtained from GenBank (Appendix A). In total, 35 sets of CDS of 17 liverworts and 18 mosses were taken for the analyses. In order to predict C-to-U and U-to-C RNA editing sites, the PREPACT 3.12.0 (Universität Bonn, Bonn, Germany) [39] tool was used with the BLASTX mode and 0.001 e-value cut-off. The PREPACT 3.12.0 (Universität Bonn, Bonn, Germany) reference database contains only one representative of liverworts and one of mosses, although these species were experimentally proven to present little (*Physcomitrella patens*—11 RNA editing sites identified) or no (*Marchantia paleacea*) evidence of RNA editing, which is desired while predicting RNA editing events. The *M. paleacea* mitogenome was used as a reference mitogenome for the liverwort CDS set analysis while the *P. patens* mitogenome was used as the reference mitogenome for the moss CDS set analysis. The prediction of RNA editing sites produced 35 strings of numbers indicating the positions of single bases within 33 genes (each such set, i.e., a string of numbers, was composed of 33 short strings of positions of bases within genes ordered in the same manner). Next, based on the strings, the number of RNA editing sites predicted in each gene of every species was calculated (Appendix A). In order to compare RNA editing across species, the effect of different gene lengths across species had to be compensated. Therefore, the ratio of the number of predicted RNA editing sites within a gene CDS to gene CDS length was expressed as a percentage and referred to as ‘editing frequency’ in this paper (Appendix A). For purposes of comparison between liverworts and mosses, the mean C-to-U RNA editing frequencies of genes of each group were calculated based on the C-to-U editing frequencies of particular genes within each group. The means of C-to-U editing frequency were scaled based on the average level of editing frequency. Since the level was 3.57 times higher in liverworts, the means of C-to-U editing frequency of mosses were multiplied by 3.57 in order to compare editing frequency distribution across genes of bryophytes.

The 1000 plants project (1KP) [50] data was used to verify predicted RNA editing sites. Out of 35 investigated species of liverworts and mosses, four were found within 1KP transcriptome database, i.e., one liverwort (*Pallavicinia lyellii*) and three mosses (*Atrichum angustatum*, *Buxbaumia aphylla*, and *Tetraphis pellucida*). The transcriptome datasets were mapped to 33 corresponding protein-coding sequences of mitochondrial genes using Geneious 8.1 software (Biomatters Ltd., Auckland, New Zealand) [69]. The obtained gene transcripts were aligned with protein-coding sequences of genes using MAFFT 7.427 (CBRC, Tokyo, Japan) [70] and the alignment mismatches were subsequently compared with the RNA editing sites predicted by PREPACT 3.12.0 [39] (Appendix A).

### 3.4. Phylogenetic Analyses

Phylogenetic analyses were performed using 37 sets of the 33 aforementioned protein-coding sequences. The sequence sets were first aligned and concatenated using Geneious 8.1 (Biomatters Ltd., Auckland, New Zealand) [69] and MAFFT 7.427 (CBRC, Tokyo, Japan) [70] and PartitionFinder2 2.1.1 (Australian National University, Canberra, Australia) [71] was then used to determine the best partitioning schemes and corresponding nucleotide substitution models (Appendix A). The dataset blocks were predefined based on protein-coding genes. Afterwards, based on the alignment and partitioning schemes, Bayesian analysis was conducted using MrBayes 3.2.1 (University of Rochester, Rochester, NY, USA) [72]. The MCMC algorithm was run for 20,000,000 generations (sampling every 1000) with four incrementally-heated chains. The first 25% trees were discarded as burn-in. The remaining trees were used to generate the consensus tree. Additionally, the Maximum Likelihood phylogenetic analysis was conducted using RAxML 8.2.11 (The Exelixis Lab, Heidelberg, Germany) [73] plugin for Geneious 8.1 (Biomatters Ltd., Auckland, New Zealand) [69] with GTR + G model and 1000 bootstrap replicates (Appendix A).

Alongside analyses based on protein-coding sequences, a phylogenetic reconstruction was also performed using the predicted RNA editing data to test the hypothesis that the distribution of RNA editing sites follows the branching pattern reflected in protein-coding sequence-based phylogeny. However, due to the use of different reference mitogenomes during the prediction of RNA editing sites, the analyses were conducted separately for liverworts and mosses. Additionally, the mitogenomes of *M. paleacea* and *P. patens* could not be included in this analysis since they were used as references in RNA editing sites predictions. First, since RNA editing sites of each species were predicted relatively to the same set of reference genes, all C-to-U editing positions identified within each gene across species were summarized as an RNA editing template of a gene. Next, the gene templates were concatenated in order to obtain one template for a whole set of genes. Afterwards, RNA editing positions of each species were mapped to the template to obtain a binary matrix of RNA editing predictions. The liverworts’ matrix contained a total of 1285 unique predicted editing sites positions, while the mosses’ matrix contained 435 (Appendix A). All analysis steps mentioned above were performed using a custom Python script (https://github.com/gymnomitrion/binary_editing.git, published and accessed on: 20 February 2019). Finally, based on a binary matrix of RNA editing converted to NEXUS file format, the Bayesian inference (20,000,000 generations with sampling every 1000) (Appendix A), maximum parsimony and neighbor-joining analyses (with 1000 bootstrap replicates and 50% cut-off) were conducted using PAUP software (https://paup.phylosolutions.com/, accessed on: 25 March 2019) and MrBayes 3.2.1 (University of Rochester, Rochester, NY, USA) [72]. The liverworts’ tree was depicted with *H. hookeri* as an outgroup, while the mosses’ tree was depicted with *S. palustre* as an outgroup (Figure 1).

### 3.5. Statistical Analyses

The Kruskal–Wallis test (*p*-value cut-off < 0.05) was used to compare the C-to-U and U-to-C means of editing frequency of genes among three groups of moss as well as three groups of liverwort. *P. patens* was excluded from the analysis, unlike *M. paleacea* (confirmed non-editing mitogenome), since previous studies reported 11 editing sites identified within its mitogenome. The post-hoc Dunn’s test with the Benjamini-Hochberg *p*-value adjustment method (*p*-value cut-off < 0.05) was used to pinpoint which means of editing frequencies were significantly different from the others. The Pearson correlation coefficient was applied to measure the correlation between means of C-to-U editing frequency. All statistical analyses were conducted using the R software environment (https://www.R-project.org/, accessed on: 27 March 2019) and PMCMR software package [74].

## 4. Conclusions

To obtain a comprehensive view of the RNA editing potential of mitogenomes of bryophytes, 37 sets of 33 mitochondrial protein-coding sequences were analyzed. The results obtained in this study concerned the main lineages of liverworts and mosses, which is the first case of such extensive study of mitogenomic RNA editing in early land plants. For the purpose of this research, the number of assembled liverwort mitochondrial genomes was expanded by seven and additionally provided two nearly complete sets of mitochondrial genes of liverworts. The newly-assembled mitogenomes display features typical of liverworts. The predictions of RNA editing sites within whole set of mitogenomes of 17 liverworts and 18 mosses species revealed that the level of C-to-U RNA editing in liverworts and mosses is group-specific. Among all investigated species, the highest number of editing sites was predicted in *H. hookeri* liverwort (712 C-to-U editing sites) and *P. cygnisetum* moss (124 C-to-U editing sites). The average mitochondrial C-to-U RNA editing level was also over three times higher in liverworts than in mosses. Despite considerable differences in editing level, the C-to-U editing frequency of majority of genes seems to be similar for each gene across the two groups of early land plants. Such genes as *atp9*, *sdh3*, or *tatC* showed high, while *atp8*, *rpl16*, and *rps12* showed low C-to-U editing frequency, both in mosses and liverworts relatively to the overall predicted C-to-U editing level. Contrarily, the U-to-C RNA editing site predictions did not reveal significant differences in editing frequency across taxonomic groups of either liverworts or mosses. However, the validation of U-to-C RNA editing predictions using database-derived transcriptomes suggests that the U-to-C prediction models require further improvement.

The analyses of RNA editing patterns demonstrated that RNA editing potential hidden in the mitochondrial genomes of liverworts and mosses is widespread and may have considerable impact on mature transcripts encoded by mitochondrial DNA, especially among liverworts. 

## Figures and Tables

**Figure 1 ijms-20-02963-f001:**
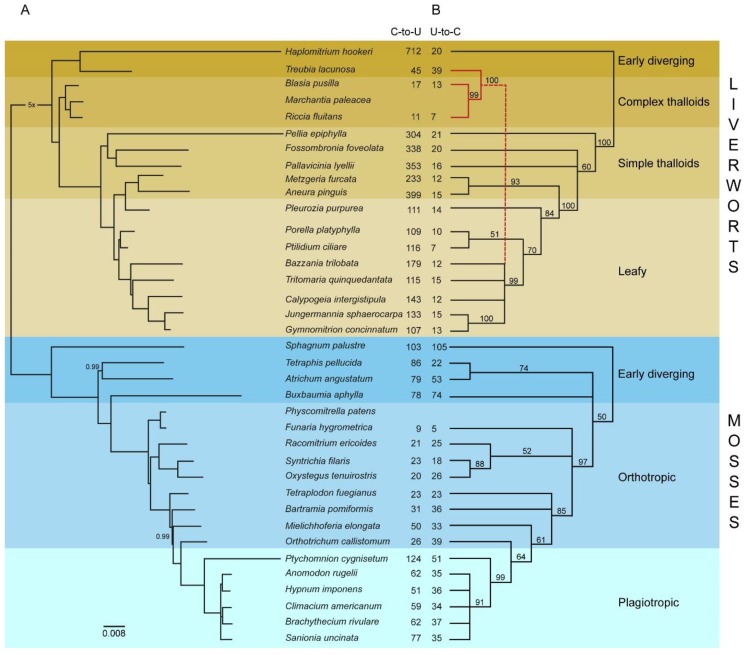
Phylogenetic relationships of 37 species of liverworts and mosses based on 33 concatenated mitochondrial protein-coding sequences and predicted RNA editing sites. (**A**) Phylogenetic tree obtained as a result of Bayesian inference analysis of 33 concatenated protein-coding sequences of 37 species. All branches are maximally supported unless otherwise marked. The scale bar indicates the number of substitutions per nucleotide position. (**B**) Phylogenetic tree obtained as a result of maximum parsimony analysis of binary matrix of predicted C-to-U RNA editing site occurrence within the 33 aforementioned protein-coding sequences of 35 species. The support values are given at the nodes. Two columns in the middle of the graph depict the total number of C-to-U and U-to-C editing sites predicted within the aforementioned species sequences.

**Figure 2 ijms-20-02963-f002:**
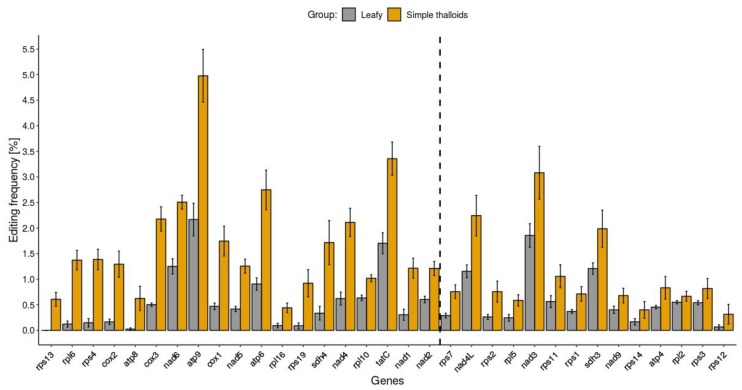
Predicted C-to-U editing frequency of liverwort genes. Bar plots depicting mean editing frequency of leafy (*n* = 8) and simple thalloid (*n* = 5) liverworts are colored by liverwort group. Genes are sorted in ascending order (left to right) of statistical significance of differences between editing frequencies. The significantly different (*p*-value < 0.05) editing frequencies are depicted on the left side of vertical dashed line. The bar plot whiskers depict standard error values.

**Figure 3 ijms-20-02963-f003:**
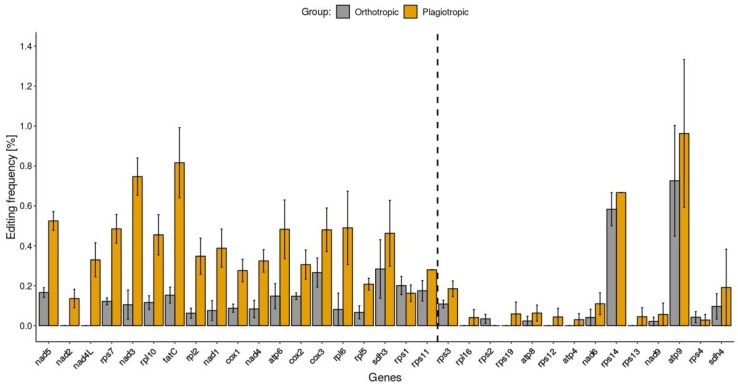
Predicted C-to-U editing frequency of moss genes. Bar plots depicting mean editing frequency of orthotropic (*n* = 8) and plagiotropic (*n* = 6) mosses are colored by moss group. Genes are sorted in ascending order (left to right) of statistical significance of differences between editing frequencies. The significantly different (*p*-value < 0.05) editing frequencies are depicted on the left side of the vertical dashed line. The bar plot whiskers depict standard error values.

**Figure 4 ijms-20-02963-f004:**
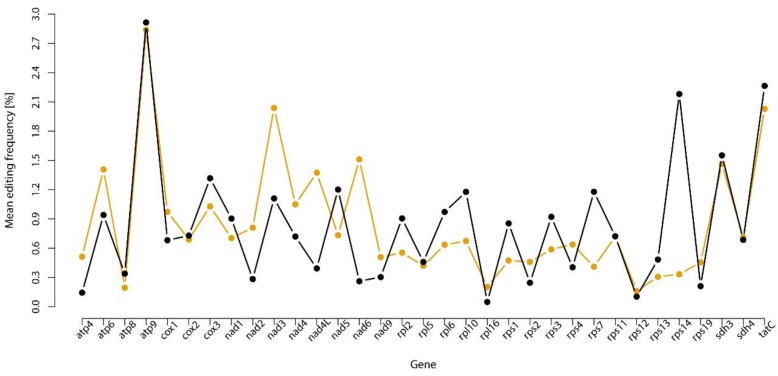
Mean C-to-U RNA editing frequency in genes of mosses and liverworts. The means of C-to-U RNA editing frequency of liverworts (orange line) and mosses (black line) are scaled.

**Table 1 ijms-20-02963-t001:** Details of the complete mitochondrial genomes of seven newly-sequenced liverworts species

Species	Accession Number	Genome Size (bp)	Mean Coverage	GC Content (%)	Total Number of Genes	Protein Coding Genes	rRNA	tRNA	Genes with Introns	Total Number of Introns
*Fossombronia foveolata*	MK749462	175,222	80.5	45.2	73	42	3	27	14	25
*Haplomitrium hookeri*	MK749465	174,448	44.4	47.6	74	43	3	27	16	30
*Jungermannia sphaerocarpa*	MK749463	156,050	64.8	44.8	72	42	3	27	14	25
*Metzgeria furcata*	MK749464	171,891	99.2	46.7	74	42	3	28	15	29
*Porella platyphylla*	MK749461	179,081	31.9	44.7	74	42	3	28	15	29
*Ptilidium ciliare*	MK749460	179,562	233.0	44.8	74	42	3	28	15	29
*Riccia fluitans*	MK749459	185,621	202.8	42.4	74	42	3	28	16	30

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
