# Peer review of "Potential of Transcript Editing Across Mitogenomes of Early Land Plants Shows Novel and Familiar Trends"

_ijms, 2019, doi:10.3390/ijms20122963_

Round 1

Reviewer 1 Report

Myszczyński and colleagues looked into (predicted) RNA editing for 33 mitochondrial genes of 36 bryophytes (18 liverworts and 18 mosses). Seven complete liverwort mitogenomes of diverse genera, largely representing interesting taxonomic groups not covered previously, were newly determined and complete [?? see below] sets of mitochondrial protein coding sequences were determined for two additional liverworts, likewise representing interesting genera (Pellia and Pallavicinia). Although not delivering fundamentally novel insights, this is a valuable study on the (predicted) occurrence of RNA editing in liverworts and mosses, the two likely most ancient clades of land plants. Although I understand that the manuscript is intended for a special issue on RNA editing, the authors could (and should!) actually emphasize that seven complete new mitogenomes have been obtained. On the other hand, they should somewhat tone down the statements on RNA editing analyses since they used a bioinformatic tool for prediction without confirmation of RNA editing on cDNA (except for some limited data retrieved from the OneKP project).

Note: The newly assembled mitogenome sequences were not yet publicly available for review and testing and we have to trust the data and insights as presented by the authors.

Comments:

1.    It should be clearly stated in the abstract and again at the beginning of results that RNA editing sites were PREDICTED, not determined or confirmed by parallel cDNA work.

2.    The list of (33) CDS included in their analyses is given in line 93 behind the wording “include such genes as”. What does that mean and what about generally conserved plant mitochondrial protein-coding genes not included in their data set: atp1, cob, rps8, rps10 and the suite of ccm (or alternatively ccb) genes encoding components of the cytochrome c biogenesis machinery? The latter are a particularly interesting case given their previously reported disintegration in Treubia. Are they degenerated or entirely absent in Haplomitrium, too? At least atp1 and cob should be universally conserved, however.

3.    Figure 1 nicely juxtaposes a molecular phylogeny (A) with a parsimony analyses of the RNA editing site matrix (B). Although detailed under methods, the procedure to obtain the tree shown under B (bootstrap consensus at 50% cutoff?) should be explicitly stated.

4.    Most importantly the RNA editing site matrix should also be made available for independent testing by the readers in the supplementary data collection!

5.    The sentence “RNA editing is not only process of increasing protein diversity, but also few genes are known of obligatory RNA edition [1]” is incomprehensible to me. Moreover, increasing proteins diversity is not confidently demonstrated for the plant-type of RNA editing under consideration here.

6.    The PREPACT tool has been used in BLASTX mode and with only Marchantia or alternatively Physcomitrella used as a reference. The authors may wish to comment on that. Wouldn’t actually (i) using the alignment prediction mode and (ii) using multiple references and he commons threshold criteria would have been more suitable for this project?

7.    In contrast to the previously determined collections of mt CDS for Bazzania and Blasia, only single database accession numbers are given for Pallavicinia and Pellia in table 1 although their mitogenomes were not completely assembled either. How can that be?

8.    Is Jungermannia sphaerocarpa identical to Solenostoma sphaerocarpum (in the NCBI Taxonomy database)?

9.    Figure legends 2 and 3 should give all details on graphic elements (dots, lines/bars)

10. Language, grammar and wording should be re-inspected, ideally by a native speaker. Just some exemplary issues from the first pages:

a.      I suggest avoiding using (RNA) “edition” instead of (RNA) editing since the former is restricted in meaning along the sense of serial emissions, issues or versions.

b.      Behind colon in line 60: ARE differences … significant?

c.      Check for correct numerus throughout the manuscript: RNA sequenceS (l.9), organismS (l.24), THESE (l.43) et c. et c.

d.      Check for the proper use of articles throughout the manuscript: THE genome (l.26), A main role (l.37/38), mitogenomeS (l.47) et c. et c.

e.      Have acronyms spelled-out at least once upon first use (e.g. PPR)

Author Response

It should be clearly stated in the abstract and again at the beginning of results that RNA editing sites were PREDICTED, not determined or confirmed by parallel cDNA work.

Corrected.

2.    The list of (33) CDS included in their analyses is given in line 93 behind the wording “include such genes as”. What does that mean and what about generally conserved plant mitochondrial protein-coding genes not included in their data set: atp1, cob, rps8, rps10 and the suite of ccm (or alternatively ccb) genes encoding components of the cytochrome c biogenesis machinery? The latter are a particularly interesting case given their previously reported disintegration in Treubia. Are they degenerated or entirely absent in Haplomitrium, too? At least atp1 and cob should be universally conserved, however.

The analyses were limited to only protein-coding genes commonly occurring in both liverwort and moss mitochondria. Other protein-coding genes were also identified, however not published in this paper.

3.    Figure 1 nicely juxtaposes a molecular phylogeny (A) with a parsimony analyses of the RNA editing site matrix (B). Although detailed under methods, the procedure to obtain the tree shown under B (bootstrap consensus at 50% cutoff?) should be explicitly stated.

Corrected.

4.    Most importantly the RNA editing site matrix should also be made available for independent testing by the readers in the supplementary data collection!

Corrected. The table is included in supplementary data (Table S9 and Table S10)

5.    The sentence “RNA editing is not only process of increasing protein diversity, but also few genes are known of obligatory RNA edition [1]” is incomprehensible to me. Moreover, increasing proteins diversity is not confidently demonstrated for the plant-type of RNA editing under consideration here.

The reference supporting the sentence was added, since this part of introduction concerns RNA editing across various species.

6.    The PREPACT tool has been used in BLASTX mode and with only Marchantia or alternatively Physcomitrella used as a reference. The authors may wish to comment on that. Wouldn’t actually (i) using the alignment prediction mode and (ii) using multiple references and he commons threshold criteria would have been more suitable for this project?

Our previous experiences with the use of PREPACT indicated that BLASTX mode provides less biases in RNA editing sites predictions. The explanation on the (ii) is now provided within main text of manuscript (lines: 70-82).

7.    In contrast to the previously determined collections of mt CDS for Bazzania and Blasia, only single database accession numbers are given for Pallavicinia and Pellia in table 1 although their mitogenomes were not completely assembled either. How can that be?

That was overlooked. Corrected.

8.    Is Jungermannia sphaerocarpa identical to Solenostoma sphaerocarpum (in the NCBI Taxonomy database)?

It is the same species.

9.    Figure legends 2 and 3 should give all details on graphic elements (dots, lines/bars)

Figures 2 and 3 were remodeled and described more clearly.

10. Language, grammar and wording should be re-inspected, ideally by a native speaker. Just some exemplary issues from the first pages:

a.         I suggest avoiding using (RNA) “edition” instead of (RNA) editing since the former is restricted in meaning along the sense of serial emissions, issues or versions.

b.         Behind colon in line 60: ARE differences … significant?

c.         Check for correct numerus throughout the manuscript: RNA sequenceS (l.9), organismS (l.24), THESE (l.43) et c. et c.

d.         Check for the proper use of articles throughout the manuscript: THE genome (l.26), A main role (l.37/38), mitogenomeS (l.47) et c. et c.

e.         Have acronyms spelled-out at least once upon first use (e.g. PPR)

Corrected.

Reviewer 2 Report

RNA editing is a broad term encompassing a range of processes that result in the production of transcripts that differ in defined ways from the genome. In plants, these changes are limited to mitochondrial and plastid transcripts and involve C-to-U and, less frequently, U-to-C conversions.  While editing has been studied extensively in flowering plants, less is known about the extent of editing in early land plants.  This manuscript by Myszczynski et al. describes an analysis of the mitochondrial genomes of 37 species of liverworts and mosses.  Existing Prepact 3.0 software was used to predict sites of C-to-U and U-to-C editing in 33 protein coding genes from each genome, which were then compared across species.  The accuracy of editing site predictions for 4 of these species was determined using data from the 1000 plants project (1KP).  The transcriptome data supported only 75% (246 of 326) of the predicted C-to-U sites and just 5% (3 of 57) of the predicted U-to-C sites in the 3 moss and single liverwort species for which transcript sequences were available, confounding any comparative analysis between the 37 species.

This work adds a significant amount of sequence data, including complete, or nearly complete, mitochondrial genomes from 18 liverwort and 18 moss species.  These data provide an excellent starting point from which to analyze phylogenetic relationships between species and RNA editing patterns.  However, the lack of substantial transcriptome data and the surprisingly low accuracy of the software used to predict editing sites reduces confidence in the significance of the results presented in Figures 2-4.  There are also concerns regarding how the data are presented, as described below.

Specific comments:

1.     Prior to any consideration for publication, the manuscript must be extensively copy edited to correct the multitude of grammatical errors, with special attention to subject-verb agreement and the use of articles (a, an, the, etc.).  These issues are nicely illustrated by the key questions posed in lines 59-64.

2.     Wording needs to be more precise throughout the manuscript.  For example, RNA editing does NOT occur in organellar and nuclear genomes, as indicated in the first sentence of the Introduction (it obviously affects transcripts), nor is it always post-transcriptional.

3.     All of the papers that are referenced in the Introduction are from the plant RNA editing literature despite the fact that the first few sentences refer to the overall process of editing.  Add general references after the first two sentences (lines 24 and 25).

4.     Insert the phrase “In plants” to make it clear that the rest of the introduction refers to editing in plant organelles.  In metazoans, for example, it is not true that editing ‘appears mainly in protein-coding regions of the genome’. 

5.     Results: The switch from the description of liverwort genomes to ‘RNA editing sites in liverworts’ (line 96) does not make it clear that these are predicted editing sites, rather than experimentally determined sites.  It is not until line 194 that transcriptome data are mentioned and, as presented in this six-line paragraph, it is not entirely clear that the authors are referring to their own data.  In fact, this reviewer did not recognize the significance of the values presented until arriving at line 288 in Materials and Methods (under Prediction and verification of RNA editing sites).

6.     Burying the verification data in Supplementary Table S7 is clearly inappropriate.  This table should be presented as Table 2 in the body of the manuscript, and fully discussed, as it colors the interpretation of the rest of the manuscript.

7.     The text on lines 342-345 is misleading at best.  “Contrarily, the U-to-C RNA editing sites predictions did not revealed significant differences in editing frequency across taxonomic groups of neither liverworts nor mosses.  However the validation of the U-to-C RNA editing predictions with the use of database-derived transcriptomes suggests that the U-to-C prediction models may require further improvement.”  The authors’ own data, mentioned only in passing on line 196 (only 5% of the predicted U-to-C sites were confirmed), clearly indicates that the model is not successful at predicting U-to-C sites.  Hence, any comparison between taxonomic groups is meaningless.

Author Response

Reviewer #2

1.     Prior to any consideration for publication, the manuscript must be extensively copy edited to correct the multitude of grammatical errors, with special attention to subject-verb agreement and the use of articles (a, an, the, etc.).  These issues are nicely illustrated by the key questions posed in lines 59-64.

Corrected.

2.     Wording needs to be more precise throughout the manuscript.  For example, RNA editing does NOT occur in organellar and nuclear genomes, as indicated in the first sentence of the Introduction (it obviously affects transcripts), nor is it always post-transcriptional.

Corrected.

3.     All of the papers that are referenced in the Introduction are from the plant RNA editing literature despite the fact that the first few sentences refer to the overall process of editing.  Add general references after the first two sentences (lines 24 and 25).

We have provided additional references regarding various species in the Introduction.

4.     Insert the phrase “In plants” to make it clear that the rest of the introduction refers to editing in plant organelles.  In metazoans, for example, it is not true that editing ‘appears mainly in protein-coding regions of the genome’.

Corrected.

5.     Results: The switch from the description of liverwort genomes to ‘RNA editing sites in liverworts’ (line 96) does not make it clear that these are predicted editing sites, rather than experimentally determined sites.  It is not until line 194 that transcriptome data are mentioned and, as presented in this six-line paragraph, it is not entirely clear that the authors are referring to their own data.  In fact, this reviewer did not recognize the significance of the values presented until arriving at line 288 in Materials and Methods (under Prediction and verification of RNA editing sites).

In order to highlight that the RNA editing sites were predicted we have provided additional text fragments in the Abstract (lines: 14-15), Introduction (lines: 71-83) and Results sections (e.g. headings). Also the origin of the transcriptomic data was stated more clearly (lines: 273-277).

6.     Burying the verification data in Supplementary Table S7 is clearly inappropriate.  This table should be presented as Table 2 in the body of the manuscript, and fully discussed, as it colors the interpretation of the rest of the manuscript.

Since the transcriptomic data was not obtained as a result of experimental validation of our samples, but downloaded from 1KP project database we believe that including this table in main manuscript may cause readers confusion. Further explanation is stated in lines: 273-290.

7.     The text on lines 342-345 is misleading at best.  “Contrarily, the U-to-C RNA editing sites predictions did not revealed significant differences in editing frequency across taxonomic groups of neither liverworts nor mosses.  However the validation of the U-to-C RNA editing predictions with the use of database-derived transcriptomes suggests that the U-to-C prediction models may require further improvement.”  The authors’ own data, mentioned only in passing on line 196 (only 5% of the predicted U-to-C sites were confirmed), clearly indicates that the model is not successful at predicting U-to-C sites.  Hence, any comparison between taxonomic groups is meaningless.

Corrected. We explained that issues also in lines: 273-290.

Reviewer 3 Report

Summary

The authors have attempted to provide a detailed view of RNA editing in liverworts and mosses using genomic data from 36 species. Through a comparative study of 33 genes among liverworts and mosses and the gene-wise comparisons - author proposes that overall frequency of RNA editing in liverwort and mosses is taxon-dependent and distribution of these edited sites within mitogenomes is locus-dependent. Based on these observations the authors conclude that RNA editing is widespread and may have a considerable impact on the mitochondrial transcriptome in these species.

Major Comments/Issues

1.     It's not clear how authors identified RNA-editing sites without the transcriptome data. If authors sequenced transcriptome for these species, then it needs to be mentioned and highlighted in results as well as in the methods sections.

2.     Since authors conclude that RNA editing is widespread in mitogenomes and may have a significant impact on the transcriptome, it’s essential to compare with RNA editing frequencies and distribution in genes from the nuclear genome. Is there a bias in RNA editing in mitogenomes or the observations from mitogenomes matches with rest of the (nuclear) genome?

Minor Issues

1.     The manuscript is hard to follow; there are leaps of logic and sometimes it’s left for readers to investigate what authors are trying to convey. For example:

a.     Line 131-132 – which ‘inconsistency’ authors are referring to is not clear; there seems there is a leap of logic,

b.     Line 198-99 – author’s states differences between their data and transcriptome from 1K transcriptome project but doesn’t attempt to provide any reasons

2.     Manuscript needs to be revised to fix grammatical, spelling mistakes. For example:

a.     Line 29 – ‘edition’ can be corrected to ‘editing events’

b.     Line 59 – ‘genomic-wide study’ should be corrected to ‘genome-wide’ study

Author Response

Reviewer #3

Major Comments/Issues

1.         It's not clear how authors identified RNA-editing sites without the transcriptome data. If authors sequenced transcriptome for these species, then it needs to be mentioned and highlighted in results as well as in the methods sections.

In this version of the manuscript we have clearly stated that RNA editing sites were predicted with the use of mitogenomic sequences and bioinformatic software - PREPACT 3.0 (see lines: 14-15, 83-84, 130, 139 etc, also highlighted in headings of subsections).

2.         Since authors conclude that RNA editing is widespread in mitogenomes and may have a significant impact on the transcriptome, it’s essential to compare with RNA editing frequencies and distribution in genes from the nuclear genome. Is there a bias in RNA editing in mitogenomes or the observations from mitogenomes matches with rest of the (nuclear) genome?

Experimental methods were mainly focused on capturing and sequencing organellar DNA, therefore we were not able to sufficiently assemble nuclear genes sequences with this material. Moreover public databases do not contain sequences of nuclear genomes of species that we have investigated.

Minor Issues

1.         The manuscript is hard to follow; there are leaps of logic and sometimes it’s left for readers to investigate what authors are trying to convey. For example:

a.         Line 131-132 – which ‘inconsistency’ authors are referring to is not clear; there seems there is a leap of logic,

b.         Line 198-99 – author’s states differences between their data and transcriptome from 1K transcriptome project but doesn’t attempt to provide any reasons

2.         Manuscript needs to be revised to fix grammatical, spelling mistakes. For example:

a.         Line 29 – ‘edition’ can be corrected to ‘editing events’

b.         Line 59 – ‘genomic-wide study’ should be corrected to ‘genome-wide’ study

Corrected.

Reviewer 4 Report

In their manuscript, Myszczyński et al. performed a comprehensive prediction of RNA editing sites across mitochondrial protein-coding genes of 37 early land plants. They analyzed how these editing sites are distributed along the phylogeny of the liverworts and mosses, and then tried to discern evolutionary patterns associated with the occurrence of RNA editing sites. The study is certainly important to get a better picture of the evolution of RNA editing, but the current manuscript lacks in several aspects, which I try to point out below. Overall, one cannot validate the study’s conclusions based on the presented results, mainly because the methodology section lacks substantial details on the calculation of the various values related to RNA editing.

General comments to the manuscript

1) The authors should avoid vague and general statements, which are frequent in some sections of the text. (See below for the details.)

2) The questions proposed in the study are arguably intriguing, but, unfortunately, the authors present the answers in a rather vague manner. This could be improved by specifying what the observed trends are (as part of the results) and by elaborating on the possible evolutionary and/or mechanistic implications of these observations (as part of the discussion, which is currently underdeveloped).

3) The methodology of the principal section of the work (i.e., RNA editing analyses) is not presented in sufficient detail. For example, it is unclear which of the predicted RNA editing sites were taken into account at various analysis steps and what were the criteria for their selection.

Abstract

(page 1, lines 14-19) The last three sentences of the abstract would benefit from some rephrasing. For example, it would be useful to specify right away what the topic of this extensive research was. Next, although the authors allude to “mitogenomic RNA editing potential”, open disclosure of the fact that RNA editing sites were for the most part predicted, as opposed to experimentally determined, would be preferable. (Even though some sites in some species were validated.) Lastly, the closing sentence of the abstract is too vague. Please elaborate; explicitly state what was newly discovered. For example, it is not clear what the locus-dependent patterns are. (The latter actually isn’t even specified in the main text of the manuscript, so is anyway unacceptable.)

Introduction

General comments

1) References in the first paragraph are rather plant-centric, somewhat ignoring the vast knowledge on the RNA editing in other groups of organisms (animals, fungi, a huge number of protist lineages). The authors should either broaden their selection of references, or explicitly state that they are discussing only the aspects of RNA editing in plants. The latter would be totally adequate for the scope of the manuscript, but needs to be explicitly acknowledged.

2) Updating a few of the broad-topic references in the Introduction would be beneficial. (Several are older than 10 years, though newer discussions and insights on the topics have been published.)

3) While it is important to discuss RNA editing for this manuscript, it seems equally important, at least briefly, to present the phylogeny of plants. This seems particularly crucial for a non-specialist reader.

Specific comments

(1, 24-25) The statement is not entirely true, as it mixes up the roles of RNA editing in nucleus and in organelles. Also, the term is “RNA editing”, not “RNA edition” (which is a repetitive error in the text). The reference #1 cites a paper on PPR proteins, which is not appropriate for the topic. Several relevant recent reviews and hypothesis articles discuss the functional and evolutionary aspects and implications of RNA editing.

(1, 28) Reference #4 — It would be more appropriate to use a reference more relevant to the topic, i.e., not on PPR proteins, but on the described phenomenon.

(1, 34) Reference #14 — As currently written, the statement which this reference accompanies, is not true. The cited paper only discusses angiosperms (so, not all plants) and only plastid genes (so, does not necessarily apply to mitochondrial genes, which the authors study here).

(1, 35-36) The statement is not clear and not what the referenced papers (references #15,16) suggest. Rather, there is usually more RNA editing with increased GC-content, which does not mean that loss of editing sites is correlated with decreased GC-content (for example, if the sites were not there from the beginning, one cannot say that they were lost).

(2, 59-64) The closing sentences of the abstract contain lots of grammatical errors (interrogative vs. indicative mood).

Results and Discussion

(5, 98) Marchantia polymorpha — This species does not seem to be present in the Fig.1, or rather a different species name is used. If this is due to changes in the nomenclature or because of taxonomic updates, please indicate this explicitly and use the correct species name everywhere.

(5, 100) editing sites were identified — “Identified” would have been appropriate if experimental proofs were obtained. In this case, the authors can only use “predicted”. This has to be made explicit in the text.

(5, 105) late-divergent group — In this case, “diverging” rather than “divergent”  seems appropriate. (The same term is also used in Figure 1 and in several other places in the text.)

(5, 110) Figure 1 legend — It would be preferable to specify whether the protein-coding sequences were obtained from the translation of edited sequences as predicted by PREPACT3.

(6, 116-117) Revise the sentence; its meaning is unclear (possibly due to grammar errors?).

(6, 119) The term “intron cut” is unclear, especially in the context where it is used. If the authors refer to precise and complete intron loss, there is no need to introduce neologisms.

(6, 120) was only partially confirmed as general rule — Please reformulate, this phrasing does not make much sense. (The authors probably mean that retro-processing leading to intron loss is not necessarily associated with the loss of editing sites.)

(6, 126) loss of editing sites comprise only C-to-U events — It would be beneficial it the authors commented on this interesting trend. What could the causes be?

(7, 167-168) editing frequencies within genes were, in most cases, similar — The presented data (Tables S5 and S6) do not quite confirm this. If similar is expressed in percentage difference, then a majority is achieved only if differences of almost 40% are accepted; in absolute values, it would have to be a difference of ~5 edits, but 7 vs 12 edits (for example) does not seem all that similar (to this reviewer). Since “similar” is a somewhat subjective, vague term, perhaps it would be appropriate to remove the vague statement and rather focus on the significant information (which anyway follows after this problematic sentence).

(8, 196) confirmed […] 75% of C-to-U predictions — While this information is not incorrect, Table S7 shows that the huge number of confirmed C-to-U edits in one species (90% out of 250 in Pallavicinia lyellii) introduces a severe bias in the statistics (Atrichum angustatum: 50% confirmed out of 20; Buxbaumia aphylla: 18% of 49; Tetraphis pellucida: 28% of 7). This raises important questions about the overall reliability of the predicted RNA editing sites (see also the comments to the Methods section below).

(8, 198-199) discrepancy […] may be due to various reasons — This statement is too vague. It would be preferable to specify these reasons for the observed discrepancies.

(8, 205-206) Unclear what the expression “association across genes” means. What kind of association?

(9, 214-221) Repeated text (same as lines 201-208).

(9, 221-229) This is one of the most important sections in the manuscript, but is rather confusing. The description of the observations is convoluted, so it is not clear what the actual observations are. The authors refer to some patterns, but it is not evident what these patterns are. The comparison of the two groups presented in the Figure 4 shows some differences in the editing frequency, but it isn’t ultimately very useful, since critical information is lacking: i) how were the mean editing frequencies calculated?; ii) what is the basis for the plot scaling?; iii) what is the statistical support for the plotted data? Lastly, the expression “association of liverwort and moss mitochondria” makes one think about cross-species hybridization, which surely isn’t intended to be the case here.

(9, 231-233) Unfortunately, one cannot validate this conclusion based on the presented arguments.

(10, 253-254) The binary character-state analyses produce usually more reliable phylogenies — This is incorrect. Binary characters are prone to state reversal and are separated by a single type of change from an ancestral state. Therefore, they are more susceptible to convergence, if multiple changes occur, than, for example, amino acid changes; the latter contain more reliable and accurate phylogenetic signal.

(10, 256-258) While using the distribution of RNA editing sites to reconstruct phylogeny is an interesting concept, this analysis has obviously much lower resolving power than one based on protein MSA. It also ignores biases arising from biological processes that can confound the signal, such as retroposition, which could erase all editing sites in a single sweep, or such as loss of a multi-functional nuclear editing factor, which may lead to selection of secondary, compensating mutations. (For a recent example of the complexities of RNA editing site “birth and death”, see https://doi.org/10.1093/gbe/evz032 .)

Material and methods

(10, 282-284) Please explain the choice of reference species. To avoid inflating the number of editing events (i.e., false positives), a more conservative approach would seem to be more appropriate for this study, i.e., selecting multiple reference species and analyzing only editing events that are systematically identified. This seems especially critical given the substantially fluctuating editing site confirmation rates for the four species that the authors could verify using the 1KP database (see the comment regarding the line 196).

(10, 285) The authors refer to “custom Python scripts”, which is not sufficient. The codes should be made available (e.g., in GitHub) and the authors should explicitly state here what the individual scripts do.

(10, 287) The “editing frequency” is arguably the most important parameter in the manuscript, but it is not clear how it was calculated. For instance, how were the sites selected? For a site to be counted, did it have to be present, across a group, in all species or in, for example, 90% of species? Or were the editing sites aggregated disregarding conservation across species?

(11, 296) Please clarify whether the sequences derived from the fully edited transcripts (as predicted by PREPACT3) were used in the phylogenetic analysis.

(11, 300) For the phylogenetic analysis, as a complement to a Bayesian tree, a good practice is to use a classical Maximum Likelihood approach, which provides methodologically independent support values.

(11, 305) It would be helpful to explain right from the start that the authors aim to analyze whether the occurrence/distribution of RNA editing sites follows the same branching pattern as the protein sequence-based phylogeny.

(11, 315) In addition to MP and NJ approaches, it should be possible to use Bayesian methods as well (as implemented, for example, in MrBayes or Phylobayes) for the analysis of the obtained editing site matrix. Similarly to the phylogenetic tree, the independent analysis approach would strengthen the argument.

Conclusion

(12, 340) The authors conclude that they found “locus-dependent patterns of C-to-U RNA editing sites”. However, these patterns/trends are never explicitly specified or characterized, so such a conclusion cannot be currently considered valid.

Tables 1 & 2

1) While Table 2 is useful because it provides data on the newly sequenced mitochondrial genomes, Table 1 could easily be transferred to the supplements.

2) If Table 1 is moved to the supplements, please add to the current Table 2 an additional column specifying the GenBank accession number.

Figure 1

1) The colors chosen for the visualization are not very colorblind-friendly. While this is not absolutely necessary, it would be very much appreciated. (For guidelines, see for example: https://www.nature.com/articles/nmeth.1618 [should be freely available]; also, http://blogs.nature.com/methagora/2013/07/data-visualization-points-of-view.htm).

2) Current resolution is insufficient, or rather the font size of the species names and all numbers has to be increased.  It is currently unreadable. (The figures seem to be rasterized in the published PDFs of the journal, so would anyway be hardly visible.)

3) It would be useful to add outgroups to the phylogenetic analysis (such as green algae and/or other land plants).

4) Minor modification suggestion: put “Liverworts” and “Mosses” on the right side of the figure.

5) In the legend, explain what the size of the scale bar means.

Figures 2 & 3

1) With the results mainly focusing on the gene-specific significance of editing differences, it might be more relevant to plot the genes on the x-axis with boxplots of ‘simple thalloid’ vs. ‘leafy’ values next to one another. Also, separating the genes along the x-axis not in the alphabetical order, but based on an informative trait, such as, for example, the rising statistical significance, may greatly facilitate the appreciation of the results by the reader. (The same applies to Fig. 2 and 3.)

2) Please specify whether the editing represents C-to-U (as one would expect based on the main text) or both C-to-U and U-to-C editing (as one would expect based on the legends).

Figure 4

1) This figure would also really benefit from the use of a colorblind-friendly color scheme.

2) Please explain how the scaling factor was chosen. (This should be mentioned at least in the Methods).

Author Response

Reviewer #4

General comments to the manuscript

1) The authors should avoid vague and general statements, which are frequent in some sections of the text. (See below for the details.)

2) The questions proposed in the study are arguably intriguing, but, unfortunately, the authors present the answers in a rather vague manner. This could be improved by specifying what the observed trends are (as part of the results) and by elaborating on the possible evolutionary and/or mechanistic implications of these observations (as part of the discussion, which is currently underdeveloped).

3) The methodology of the principal section of the work (i.e., RNA editing analyses) is not presented in sufficient detail. For example, it is unclear which of the predicted RNA editing sites were taken into account at various analysis steps and what were the criteria for their selection.

Abstract

(page 1, lines 14-19) The last three sentences of the abstract would benefit from some rephrasing. For example, it would be useful to specify right away what the topic of this extensive research was. Next, although the authors allude to “mitogenomic RNA editing potential”, open disclosure of the fact that RNA editing sites were for the most part predicted, as opposed to experimentally determined, would be preferable. (Even though some sites in some species were validated.) Lastly, the closing sentence of the abstract is too vague. Please elaborate; explicitly state what was newly discovered. For example, it is not clear what the locus-dependent patterns are. (The latter actually isn’t even specified in the main text of the manuscript, so is anyway unacceptable.)

We have rearranged the abstract sections with consideration of above remarks.

Introduction

General comments

1) References in the first paragraph are rather plant-centric, somewhat ignoring the vast knowledge on the RNA editing in other groups of organisms (animals, fungi, a huge number of protist lineages). The authors should either broaden their selection of references, or explicitly state that they are discussing only the aspects of RNA editing in plants. The latter would be totally adequate for the scope of the manuscript, but needs to be explicitly acknowledged.

We have provided additional references regarding various species in the Introduction.

2) Updating a few of the broad-topic references in the Introduction would be beneficial. (Several are older than 10 years, though newer discussions and insights on the topics have been published.)

Corrected.

3) While it is important to discuss RNA editing for this manuscript, it seems equally important, at least briefly, to present the phylogeny of plants. This seems particularly crucial for a non-specialist reader.

We have added the information about phylogeny of bryophytes (lines: 40-52).

Specific comments

(1, 24-25) The statement is not entirely true, as it mixes up the roles of RNA editing in nucleus and in organelles. Also, the term is “RNA editing”, not “RNA edition” (which is a repetitive error in the text). The reference #1 cites a paper on PPR proteins, which is not appropriate for the topic. Several relevant recent reviews and hypothesis articles discuss the functional and evolutionary aspects and implications of RNA editing.

Corrected.

(1, 28) Reference #4 — It would be more appropriate to use a reference more relevant to the topic, i.e., not on PPR proteins, but on the described phenomenon.

Corrected.

(1, 34) Reference #14 — As currently written, the statement which this reference accompanies, is not true. The cited paper only discusses angiosperms (so, not all plants) and only plastid genes (so, does not necessarily apply to mitochondrial genes, which the authors study here).

Corrected (lines: 53-55).

(1, 35-36) The statement is not clear and not what the referenced papers (references #15,16) suggest. Rather, there is usually more RNA editing with increased GC-content, which does not mean that loss of editing sites is correlated with decreased GC-content (for example, if the sites were not there from the beginning, one cannot say that they were lost).

We have removed this text fragment.

(2, 59-64) The closing sentences of the abstract contain lots of grammatical errors (interrogative vs. indicative mood).

Corrected.

Results and Discussion

(5, 98) Marchantia polymorpha — This species does not seem to be present in the Fig.1, or rather a different species name is used. If this is due to changes in the nomenclature or because of taxonomic updates, please indicate this explicitly and use the correct species name everywhere.

Corrected.

(5, 100) editing sites were identified — “Identified” would have been appropriate if experimental proofs were obtained. In this case, the authors can only use “predicted”. This has to be made explicit in the text.

In the improved version of manuscript we have clearly stated that RNA editing sites were predicted not identified.

(5, 105) late-divergent group — In this case, “diverging” rather than “divergent” seems appropriate. (The same term is also used in Figure 1 and in several other places in the text.)

Corrected.

(5, 110) Figure 1 legend — It would be preferable to specify whether the protein-coding sequences were obtained from the translation of edited sequences as predicted by PREPACT3.

Corrected.

(6, 116-117) Revise the sentence; its meaning is unclear (possibly due to grammar errors?).

Corrected.

(6, 119) The term “intron cut” is unclear, especially in the context where it is used. If the authors refer to precise and complete intron loss, there is no need to introduce neologisms.

Corrected.

 (6, 120) was only partially confirmed as general rule — Please reformulate, this phrasing does not make much sense. (The authors probably mean that retro-processing leading to intron loss is not necessarily associated with the loss of editing sites.)

Rephrased.

(6, 126) loss of editing sites comprise only C-to-U events — It would be beneficial it the authors commented on this interesting trend. What could the causes be?

This may suggest different functional relevance of both types of editing or differences in accuracy of bioinformatic predictions between C-to-U and U-to-C editing.

(7, 167-168) editing frequencies within genes were, in most cases, similar — The presented data (Tables S5 and S6) do not quite confirm this. If similar is expressed in percentage difference, then a majority is achieved only if differences of almost 40% are accepted; in absolute values, it would have to be a difference of ~5 edits, but 7 vs 12 edits (for example) does not seem all that similar (to this reviewer). Since “similar” is a somewhat subjective, vague term, perhaps it would be appropriate to remove the vague statement and rather focus on the significant information (which anyway follows after this problematic sentence).

We have removed this sentence.

(8, 196) confirmed […] 75% of C-to-U predictions — While this information is not incorrect, Table S7 shows that the huge number of confirmed C-to-U edits in one species (90% out of 250 in Pallavicinia lyellii) introduces a severe bias in the statistics (Atrichum angustatum: 50% confirmed out of 20; Buxbaumia aphylla: 18% of 49; Tetraphis pellucida: 28% of 7). This raises important questions about the overall reliability of the predicted RNA editing sites (see also the comments to the Methods section below).

We have clearly stated the ratios of confirmed to predicted edits (lines: 276-278).

(8, 198-199) discrepancy […] may be due to various reasons — This statement is too vague. It would be preferable to specify these reasons for the observed discrepancies.

We have provided explanation on possible discrepancies (lines: 274-290).

(8, 205-206) Unclear what the expression “association across genes” means. What kind of association?

Rephrased.

(9, 214-221) Repeated text (same as lines 201-208).

Removed.

(9, 221-229) This is one of the most important sections in the manuscript, but is rather confusing. The description of the observations is convoluted, so it is not clear what the actual observations are. The authors refer to some patterns, but it is not evident what these patterns are. The comparison of the two groups presented in the Figure 4 shows some differences in the editing frequency, but it isn’t ultimately very useful, since critical information is lacking: i) how were the mean editing frequencies calculated?; ii) what is the basis for the plot scaling?; iii) what is the statistical support for the plotted data? Lastly, the expression “association of liverwort and moss mitochondria” makes one think about cross-species hybridization, which surely isn’t intended to be the case here.

We have rearranged this whole subsection (lines: 292-339) providing additional informations and clarification.

(9, 231-233) Unfortunately, one cannot validate this conclusion based on the presented arguments.

Removed.

(10, 253-254) The binary character-state analyses produce usually more reliable phylogenies — This is incorrect. Binary characters are prone to state reversal and are separated by a single type of change from an ancestral state. Therefore, they are more susceptible to convergence, if multiple changes occur, than, for example, amino acid changes; the latter contain more reliable and accurate phylogenetic signal.

Rephrased.

(10, 256-258) While using the distribution of RNA editing sites to reconstruct phylogeny is an interesting concept, this analysis has obviously much lower resolving power than one based on protein MSA. It also ignores biases arising from biological processes that can confound the signal, such as retroposition, which could erase all editing sites in a single sweep, or such as loss of a multi-functional nuclear editing factor, which may lead to selection of secondary, compensating mutations. (For a recent example of the complexities of RNA editing site “birth and death”, see https://doi.org/10.1093/gbe/evz032 .)

We were aware of such bias connected with the use of binary data, however our intention was to investigate conservation of RNA editing sites distribution rather than consequences of RNA editing.

Material and methods

(10, 282-284) Please explain the choice of reference species. To avoid inflating the number of editing events (i.e., false positives), a more conservative approach would seem to be more appropriate for this study, i.e., selecting multiple reference species and analyzing only editing events that are systematically identified. This seems especially critical given the substantially fluctuating editing site confirmation rates for the four species that the authors could verify using the 1KP database (see the comment regarding the line 196).

The explanation on the this issue is now provided within main text of manuscript (lines: 71-83).

(10, 285) The authors refer to “custom Python scripts”, which is not sufficient. The codes should be made available (e.g., in GitHub) and the authors should explicitly state here what the individual scripts do.

Unfortunately the script that was used in the study is still in developmental stage. The code is not written in user-friendly manner and further improvements are required to publish this draft version. Hence we highlighted in the manuscript that used script is "unpublished".

(10, 287) The “editing frequency” is arguably the most important parameter in the manuscript, but it is not clear how it was calculated. For instance, how were the sites selected? For a site to be counted, did it have to be present, across a group, in all species or in, for example, 90% of species? Or were the editing sites aggregated disregarding conservation across species?

The ‘editing frequency’ calculation is now described clearly in ‘Prediction and verification of RNA editing sites’ subsection.

(11, 296) Please clarify whether the sequences derived from the fully edited transcripts (as predicted by PREPACT3) were used in the phylogenetic analysis.

They were not. We have used only binary-state matrices to construct phylogenetic tree (Figure 1B). We have clarified this in lines: 428-438 (Materials and Methods).

(11, 300) For the phylogenetic analysis, as a complement to a Bayesian tree, a good practice is to use a classical Maximum Likelihood approach, which provides methodologically independent support values.

We have generated ML phylogenetic analysis as suggested and included the outcome in supplementary materials.

(11, 305) It would be helpful to explain right from the start that the authors aim to analyze whether the occurrence/distribution of RNA editing sites follows the same branching pattern as the protein sequence-based phylogeny.

Following the reviewer suggestion we have included this information.

(11, 315) In addition to MP and NJ approaches, it should be possible to use Bayesian methods as well (as implemented, for example, in MrBayes or Phylobayes) for the analysis of the obtained editing site matrix. Similarly to the phylogenetic tree, the independent analysis approach would strengthen the argument.

We have generated Bayesian phylogenetic analysis as suggested and included the outcome in supplementary materials.

Conclusion

(12, 340) The authors conclude that they found “locus-dependent patterns of C-to-U RNA editing sites”. However, these patterns/trends are never explicitly specified or characterized, so such a conclusion cannot be currently considered valid.

This statement has been removed from the manuscript text 

Tables 1 & 2

1) While Table 2 is useful because it provides data on the newly sequenced mitochondrial genomes, Table 1 could easily be transferred to the supplements.

Table 1 is now included as part of supplementary materials.

2) If Table 1 is moved to the supplements, please add to the current Table 2 an additional column specifying the GenBank accession number.

Corrected.

Figure 1

1) The colors chosen for the visualization are not very colorblind-friendly. While this is not absolutely necessary, it would be very much appreciated. (For guidelines, see for example: https://www.nature.com/articles/nmeth.1618 [should be freely available]; also, http://blogs.nature.com/methagora/2013/07/data-visualization-points-of-view.htm).

Corrected.

2) Current resolution is insufficient, or rather the font size of the species names and all numbers has to be increased.  It is currently unreadable. (The figures seem to be rasterized in the published PDFs of the journal, so would anyway be hardly visible.)

Along with the manuscript file we have separately provided figure files created as vector graphics.

3) It would be useful to add outgroups to the phylogenetic analysis (such as green algae and/or other land plants).

Since the phylogenetic analysis was conducted on particular set of mitochondrial genes adding outgroups like Chara will require usage of unified set of genes and cause reduction of the set to ca. 20 mitochondrial genes.

4) Minor modification suggestion: put “Liverworts” and “Mosses” on the right side of the figure.

Corrected.

5) In the legend, explain what the size of the scale bar means.

Corrected.

Figures 2 & 3

1) With the results mainly focusing on the gene-specific significance of editing differences, it might be more relevant to plot the genes on the x-axis with boxplots of ‘simple thalloid’ vs. ‘leafy’ values next to one another. Also, separating the genes along the x-axis not in the alphabetical order, but based on an informative trait, such as, for example, the rising statistical significance, may greatly facilitate the appreciation of the results by the reader. (The same applies to Fig. 2 and 3.)

According to the suggestion we have remodeled the figures.

2) Please specify whether the editing represents C-to-U (as one would expect based on the main text) or both C-to-U and U-to-C editing (as one would expect based on the legends).

Corrected.

Figure 4

1) This figure would also really benefit from the use of a colorblind-friendly color scheme.

Corrected.

2) Please explain how the scaling factor was chosen. (This should be mentioned at least in the Methods).

Corrected (lines: 401-403).

Round 2

Reviewer 2 Report

This revised manuscript is greatly improved over the original submission. However, additional revisions would be necessary prior to consideration for publication.

Specific comments:

1.     Prior to any consideration for publication, the manuscript must be extensively copy edited to correct the multitude of grammatical errors, with special attention to subject-verb agreement and the use of articles (a, an, the, etc.). These issues are nicely illustrated by the key questions posed in lines 59-64.

This has NOT been addressed.  The paper MUST be extensively copy edited by a native English speaker.

2.     Wording needs to be more precise throughout the manuscript.  For example, RNA editing does NOT occur in organellar and nuclear genomes, as indicated in the first sentence of the Introduction (it obviously affects transcripts), nor is it always post-transcriptional.  

Wording awkward but acceptable.

3.     All of the papers that are referenced in the Introduction are from the plant RNA editing literature despite the fact that the first few sentences refer to the overall process of editing.  Add general references after the first two sentences (lines 24 and 25).

References 1-3 are still plant-specific rather than general RNA editing references.  While the referenced papers may refer to editing in other systems, they are not adequate substitutes for inclusion of a more global review of RNA editing.  For instance, the authors must certainly be familiar with the review by Knoop in Cell. Mol. Life Sci. (2011) 68:567–586.  DOI 10.1007/s00018-010-0538-9

4.     Insert the phrase “In plants” to make it clear that the rest of the introduction refers to editing in plant organelles.  In metazoans, for example, it is not true that editing ‘appears mainly in protein-coding regions of the genome’. 

Done.

5.     Results: The switch from the description of liverwort genomes to ‘RNA editing sites in liverworts’ (line 96) does not make it clear that these are predicted editing sites, rather than experimentally determined sites.  It is not until line 194 that transcriptome data are mentioned and, as presented in this six-line paragraph, it is not entirely clear that the authors are referring to their own data.  In fact, this reviewer did not recognize the significance of the values presented until arriving at line 288 in Materials and Methods (under Prediction and verification of RNA editing sites).

6.     Burying the verification data in Supplementary Table S7 is clearly inappropriate.  This table should be presented as Table 2 in the body of the manuscript, and fully discussed, as it colors the interpretation of the rest of the manuscript.

The major concerns expressed in points 5 and 6 above have been largely addressed, but a few changes are still needed:  i. The legends to Figures 2 and 3 should read ‘PredictedC-to U editing frequency…’  ii.  ~Line 216 when discussing U-to-C predictions, it should be mentioned that even these low numbers are likely to be overestimates due to lower accuracy of the program for U-to-C sites.  iii.  Line 233 C-to-U and U-to-C editing frequencies were predicted(not calculated). OK to use calculated later in the paragraph.  iv. Lines 268-272  the comparison of the frequency of C-to-U vs U-to-C sites seems meaningless given the inaccuracy of the U-to-C predictions.  A more general statement indicating that the levels are more even would be more appropriate here.  v. line 330  the results obtained heresuggestthat atp9 (not show)

7.     The text on lines 342-345 is misleading at best.  “Contrarily, the U-to-C RNA editing sites predictions did not revealed significant differences in editing frequency across taxonomic groups of neither liverworts nor mosses.  However the validation of the U-to-C RNA editing predictions with the use of database-derived transcriptomes suggests that the U-to-C prediction models may require further improvement.”  The authors’ own data, mentioned only in passing on line 196 (only 5% of the predicted U-to-C sites were confirmed), clearly indicates that the model is not successful at predicting U-to-C sites.  Hence, any comparison between taxonomic groups is meaningless.

The expanded paragraph is an improvement over the previous version, but could be clearer. State the major finding in the first sentence:  ‘However, the U-to-C RNA editing predictions [34] should be treated with caution, as only 5% of the predicted U-to-C sites were confirmed by transcriptome data.’

Author Response

Specific comments:

1.     Prior to any consideration for publication, the manuscript must be extensively copy edited to correct the multitude of grammatical errors, with special attention to subject-verb agreement and the use of articles (a, an, the, etc.). These issues are nicely illustrated by the key questions posed in lines 59-64.

This has NOT been addressed.  The paper MUST be extensively copy edited by a native English speaker.

2.     Wording needs to be more precise throughout the manuscript.  For example, RNA editing does NOT occur in organellar and nuclear genomes, as indicated in the first sentence of the Introduction (it obviously affects transcripts), nor is it always post-transcriptional. 

Wording awkward but acceptable.

1 and 2 corrected. The manuscript was reviewed by native speaker.

3.     All of the papers that are referenced in the Introduction are from the plant RNA editing literature despite the fact that the first few sentences refer to the overall process of editing.  Add general references after the first two sentences (lines 24 and 25).

References 1-3 are still plant-specific rather than general RNA editing references.  While the referenced papers may refer to editing in other systems, they are not adequate substitutes for inclusion of a more global review of RNA editing.  For instance, the authors must certainly be familiar with the review by Knoop in Cell. Mol. Life Sci. (2011) 68:567–586.  DOI 10.1007/s00018-010-0538-9

We have accidentally included References from previous version of manuscript (containing 51 references), the proper References are included in this version of manuscript.

5.     Results: The switch from the description of liverwort genomes to ‘RNA editing sites in liverworts’ (line 96) does not make it clear that these are predicted editing sites, rather than experimentally determined sites.  It is not until line 194 that transcriptome data are mentioned and, as presented in this six-line paragraph, it is not entirely clear that the authors are referring to their own data.  In fact, this reviewer did not recognize the significance of the values presented until arriving at line 288 in Materials and Methods (under Prediction and verification of RNA editing sites).

6.     Burying the verification data in Supplementary Table S7 is clearly inappropriate.  This table should be presented as Table 2 in the body of the manuscript, and fully discussed, as it colors the interpretation of the rest of the manuscript.

The major concerns expressed in points 5 and 6 above have been largely addressed, but a few changes are still needed:  i. The legends to Figures 2 and 3 should read ‘Predicted C-to U editing frequency…’  ii.  ~Line 216 when discussing U-to-C predictions, it should be mentioned that even these low numbers are likely to be overestimates due to lower accuracy of the program for U-to-C sites.  iii.  Line 233 C-to-U and U-to-C editing frequencies were predicted(not calculated). OK to use calculated later in the paragraph.  iv. Lines 268-272  the comparison of the frequency of C-to-U vs U-to-C sites seems meaningless given the inaccuracy of the U-to-C predictions.  A more general statement indicating that the levels are more even would be more appropriate here.  v. line 330  the results obtained here suggest that atp9 (not show)

Corrected.

7.     The text on lines 342-345 is misleading at best.  “Contrarily, the U-to-C RNA editing sites predictions did not revealed significant differences in editing frequency across taxonomic groups of neither liverworts nor mosses.  However the validation of the U-to-C RNA editing predictions with the use of database-derived transcriptomes suggests that the U-to-C prediction models may require further improvement.”  The authors’ own data, mentioned only in passing on line 196 (only 5% of the predicted U-to-C sites were confirmed), clearly indicates that the model is not successful at predicting U-to-C sites.  Hence, any comparison between taxonomic groups is meaningless.

The expanded paragraph is an improvement over the previous version, but could be clearer. State the major finding in the first sentence:  ‘However, the U-to-C RNA editing predictions [34] should be treated with caution, as only 5% of the predicted U-to-C sites were confirmed by transcriptome data.’

Corrected. The cause and effect has been underlined in the first sentence of the paragraph (8, 269-274).

Reviewer 3 Report

The conclusions are mainly based on computational predictions of RNA editing sites. For a majority of species (31 out of 35) included in this study, the predictions were not verified through transcriptome sequencing, which is the current standard in RNA-editing studies, therefore expected from Authors.

In the absence of such crucial data, and given that conclusions are as reliable as the predictions -  can authors provide estimates of precision (also referred to as positive predictive value) for their predictions (from PREPACT 2.0)?  Maybe authors can use data for four species from the 1KP database, plus generate some new data for a subset of species included in this study, to compute the precision for predictions of RNA-editing sites.

Author Response

The conclusions are mainly based on computational predictions of RNA editing sites. For a majority of species (31 out of 35) included in this study, the predictions were not verified through transcriptome sequencing, which is the current standard in RNA-editing studies, therefore expected from Authors.

Despite the fact that the research focus on 35 specimens of early land plants we have sequenced mitogenomic DNA of 9 of them (the rest was database-derived). Furthermore, the majority of studied samples were collected from herbarium specimens. Therefore we were not able to extract sufficient amount of good quality RNA for transcriptome sequencing. Still we managed to quasi-verify predictions of four species.

According to previous suggestions of the other two reviewers in the manuscript we clearly state, in order to avoid readers confusion, that the presented editing potential is based on predictions. The explanation on the usage of RNA editing sites predictions was provided within main text of manuscript (2, 71-83).

In the absence of such crucial data, and given that conclusions are as reliable as the predictions -  can authors provide estimates of precision (also referred to as positive predictive value) for their predictions (from PREPACT 2.0)?  Maybe authors can use data for four species from the 1KP database, plus generate some new data for a subset of species included in this study, to compute the precision for predictions of RNA-editing sites.

We have provided such estimation in the lines: 269-274, followed by the explanation of discrepancy between our dataset and database-derived transcriptomes. Again, we could not provide new transcriptomic data of sequenced mitochondria since the majority of studied samples were collected from herbarium specimens.

Reviewer 4 Report

In their revised manuscript, Myszczyński et al. made several suggested improvements. The writing clarity is much better in the “Results and Discussion” section and the methodology information critically lacking in the previous version was amended, but further refinement is possible as detailed below. Most issues are now minor, though the authors should also take care to avoid statements, which are not substantiated by their results.

General comments

1) It is unfortunate that this manuscript version seems to have been submitted to the journal without a thorough check of the content, especially of the references. For example, the authors claim to have provided additional references regarding various species in the “Introduction” and to have updated their reference repertoire, but no such changes could be seen in the document available for the review: many references in the manuscript are off-topic or otherwise inappropriate. Virtually all references are problematic: some seem to have been shifted (referring to another topic mentioned in another sentence in that paragraph), some are completely off-topic, and refs. >51 are completely missing. Whatever the reason for these errors, the authors should make sure to carefully examine their manuscript before submission.

2) The methodology still lacks sufficient detail in several subsections. (Further details below.)

Abstract

(page 1, lines 20-21) The claimed trend of “late-diverging containing less C-to-U RNA editing than early-diverging lineages of liverworts”, which is also repeated in the “Conclusion” section (lines 461-462), does not hold up to scrutiny. As the Fig. 1 summarizes it, complex thalloids diverge earlier than simple thalloid species, but have much lower numbers of predicted editing sites. One would have to assume a very loose and almost self-serving definition of what early- and late-diverging lineages are to accept the assertion. The authors should either avoid it completely, or at least reformulate it, so that it is actually consistent with the presented results.

Introduction

(1, 27) “RNA editing is a transcriptional modification of organellar and nuclear genomes” — Erroneous statement, because RNA editing is not a modification of genomes, but of transcripts encoded by those genomes.

(1, 41) Ref. 21 is an example of an inappropriate reference: the article is on angiosperms, so citing it in support of RNA editing in bryophytes is off-topic/irrelevant.

(1, 44) Refs. 22-25 are examples of completely inappropriate references for the statements in the sentence.

(2, 85) The text should refer to Suppl. Table S1, not Table 1.

Results & Discussion

(5, 168-171) The authors surmise that to test the hypothesis that retro-processing is a mechanism of intron and editing site loss, more mitogenomes are needed. Claims that adding more data will help solving a question are not always substantiated. Why is such an analysis still impossible even after the authors contributed several more mitogenomes? If the authors consider the current dataset insufficient to address the question, they should explain in more detail what the limitations are. Plus, proposing how many more mitogenomes—and from which lineages—would be required would leverage the authors' reasoning.

(5, 171-172) The authors observe that only C-to-U, but not U-to-C editing site frequency decrease in certain lineages. In addition to two reasons for this observation that the authors provide, another that comes to mind is that the retro-processing does not actually play a role. To differentiate among the three possibilities, it could be useful to make a comparison of distribution of predicted C-to-U and U-to-C editing sites across individual genes (is there a pattern in the distribution of the predicted sites across genes?). One can probably obtain that information from the supplementary tables, but showing that directly in the manuscript (preferably in a graphical form) would make more sense.

(10, 345-346) “The mitogenomic analysis confirmed paraphylly on simple thalloid 345 liverworts and resolved enigmatic leafy genus Pleurozia as a sister to Metzgeriales.” — One would expect that the editing site frequency is linked to mt-gene sequences rather than to the body morphology. Do the authors have an explanation for this discrepancy, which one can readily see for example in the branch leading to M. fulcata, A. pinguis, and P. purpurea (Figure 1), which does not follow the morphological sorting?

Methods

(11, 412-413) Based on the authors’ reply to the previous query, it seems that, for the phylogeny inference, they used sequences encoded by the mitogenomes, i.e., in their pre-editing state. This is a questionable choice, as the sequences prior to editing are non-functional. Using the (predicted) edited transcripts sequences or protein sequences (obtained from edited transcript sequences) would have been more reliable. At the very least, a comparison of the results between the pre-edited and edited sequences would be appropriate.

(11, 413-414) If the order of steps—concatenation, then sequence alignment—was performed as stated by the authors, the results are uncertain. Concatenation followed by alignment is prone to artifacts. First, align each gene (or protein), then concatenate. In addition, prior to the phylogeny inference, one should remove dubiously aligned positions (e.g., using trimAl, G-blocks or similar tools).

(11, 415) Please specify which model was chosen as the best fit and what was the best partitioning scheme.

(11, 418) The burn-in is better expressed as a percentage of the produced trees.

(11, 420) How were the ML analyses performed? Which algorithm and parameters? (In addition, Geneious does not provide a native ML tree-generation tool, but plugins to run RAxML, PhyML, or Fast-Tree.)

(11, 421) Figure S1 — Even if the topology is the same, it would be nice if the tree was provided with the same species arrangement, so that the reader does not need to puzzle too much over it to see that the branching is congruent.

(12, 435) The argument provided in the response to the reviewer that ‘the script that was used in the study is still in developmental stage’ is unacceptable. This is like arguing that one will not provide a protocol for a wet-lab experiment, but wants to publish the results based on the said protocol. The scripts should be—at the very least—made available in the supplementary information, together with the relevant descriptions (please read the journal’s policies).

Figure 1

1) For better readability, compress the ‘editing-site’ tree (on the right side) to ~2/3 and make the species names and editing site numbers (in the central part of the figure) larger.

2 ) The color scheme has not been changed at all (and thus remained colorblind unfriendly), despite authors' claim of having corrected that issue.

Figure 2 & 3

1) If they the bar plots are arranged in the order of decreasing significance, state this in the legend. (It would be helpful.)

2) Deviations going into negative values do not make much sense. (How can the deviation be larger than the number of editing sites?)

Typos and suggested text edits

(1, 19) “spann through almost” — should rather be “span almost”

(1, 29-30) “but also few genes are known of obligatory RNA editing” — ‘several’ instead of ‘few’ is more appropriate here; replace ‘known of’ with ‘known to undergo’

(2, 85) “including 18 liverworts and 18 mosses ” — ‘comprising’ would have been more appropriate (‘including’ makes it seem as if other species in addition to the 18+18 were analyzed)

(3, 97) “in the mitogenomes early land plants” — should be “in the mitogenomes of early land plants”

(5, 190) “The Fourteen” — remove ‘The’

(6, 214) “contrairly” — ‘contrarily’

(6, 220) “does not fit to the lost of editing sites” — ‘does not fit the loss of editing sites’

(7, 247) “were” — ‘where’

(11, 395) “n order” — ‘In order’

Author Response

In their revised manuscript, Myszczyński et al. made several suggested improvements. The writing clarity is much better in the “Results and Discussion” section and the methodology information critically lacking in the previous version was amended, but further refinement is possible as detailed below. Most issues are now minor, though the authors should also take care to avoid statements, which are not substantiated by their results.

General comments

1) It is unfortunate that this manuscript version seems to have been submitted to the journal without a thorough check of the content, especially of the references. For example, the authors claim to have provided additional references regarding various species in the “Introduction” and to have updated their reference repertoire, but no such changes could be seen in the document available for the review: many references in the manuscript are off-topic or otherwise inappropriate. Virtually all references are problematic: some seem to have been shifted (referring to another topic mentioned in another sentence in that paragraph), some are completely off-topic, and refs. >51 are completely missing. Whatever the reason for these errors, the authors should make sure to carefully examine their manuscript before submission.

We have accidentally included References from previous version of manuscript (containing 51 references), the proper References are included in this version of manuscript.

2) The methodology still lacks sufficient detail in several subsections. (Further details below.)

Abstract

(page 1, lines 20-21) The claimed trend of “late-diverging containing less C-to-U RNA editing than early-diverging lineages of liverworts”, which is also repeated in the “Conclusion” section (lines 461-462), does not hold up to scrutiny. As the Fig. 1 summarizes it, complex thalloids diverge earlier than simple thalloid species, but have much lower numbers of predicted editing sites. One would have to assume a very loose and almost self-serving definition of what early- and late-diverging lineages are to accept the assertion. The authors should either avoid it completely, or at least reformulate it, so that it is actually consistent with the presented results.

We have reformulated this statement and suggested that C-to-U editing is  group-specific.

Introduction

(1, 27) “RNA editing is a transcriptional modification of organellar and nuclear genomes” — Erroneous statement, because RNA editing is not a modification of genomes, but of transcripts encoded by those genomes.

Corrected.

(1, 41) Ref. 21 is an example of an inappropriate reference: the article is on angiosperms, so citing it in support of RNA editing in bryophytes is off-topic/irrelevant.

Corrected.

(1, 44) Refs. 22-25 are examples of completely inappropriate references for the statements in the sentence.

Corrected.

(2, 85) The text should refer to Suppl. Table S1, not Table 1.

Corrected.

Results & Discussion

(5, 168-171) The authors surmise that to test the hypothesis that retro-processing is a mechanism of intron and editing site loss, more mitogenomes are needed. Claims that adding more data will help solving a question are not always substantiated. Why is such an analysis still impossible even after the authors contributed several more mitogenomes? If the authors consider the current dataset insufficient to address the question, they should explain in more detail what the limitations are. Plus, proposing how many more mitogenomes—and from which lineages—would be required would leverage the authors' reasoning.

It’s difficult to propose exact numbers, because there’s no clear evolutionary pattern of intron loss, which is one of the proofs of retropocessing (and most reliable at DNA level). Generally, intron loss appears randomly in some species of leafy liverworts, but it seems to be variable even on genus level. So, if the researchers were unlucky, they can sequence many mitogenomes with stable intron number and still need to search for more.

(5, 171-172) The authors observe that only C-to-U, but not U-to-C editing site frequency decrease in certain lineages. In addition to two reasons for this observation that the authors provide, another that comes to mind is that the retro-processing does not actually play a role. To differentiate among the three possibilities, it could be useful to make a comparison of distribution of predicted C-to-U and U-to-C editing sites across individual genes (is there a pattern in the distribution of the predicted sites across genes?). One can probably obtain that information from the supplementary tables, but showing that directly in the manuscript (preferably in a graphical form) would make more sense.

Prior to submitting the manuscript to the Journal we have provided such figures. However we have decided that presenting the homogeneous distribution of predicted U-to-C editing sites across individual genes brings no new information therefore we have just stated that observation in the manuscript. We have provided the separate paragraph on the distribution of predicted editing sites across genes - “RNA editing across mitochondrial genes of bryophytes”. However it is focused on the, much more interesting, C-to-U RNA editing sites predictions since the U-to-C RNA editing sites predictions seems to be constant and less accurately predicted.

(10, 345-346) “The mitogenomic analysis confirmed paraphylly on simple thalloid 345 liverworts and resolved enigmatic leafy genus Pleurozia as a sister to Metzgeriales.” — One would expect that the editing site frequency is linked to mt-gene sequences rather than to the body morphology. Do the authors have an explanation for this discrepancy, which one can readily see for example in the branch leading to M. fulcata, A. pinguis, and P. purpurea (Figure 1), which does not follow the morphological sorting?

Pleuroziales are phylogenetically  quite isolated from other hepatics. Although characterized by well-differentiated stems and leaves, Pleuroziales are placed in the Metzgeriidae based on molecular data (Crandall-Stotler et al. 2000 - Morphology and classification of the Marchantiophyta. In: SHAW, A. J. & GOFFINET, B. (eds) Bryophyte Biology, pp. 21–70., Crandall-Stotler et al. 2009 doi:10.1017/S0960428609005393), as sister group to the rest of the simple thalloid hepatics. Our results based on nucleotide sequences of mitochondrial genes are consistent with previous study, but the distribution of editing sites revealed, for the first time, the monophyly of leafy liverworts. We do not think that proteins derived from mitogenome transcripts play any role in morphology, but editing generally restores ancestral like protein (Sloan 2017 - https://doi.org/10.1098/rsbl.2017.0314), so therefore the leafy liverworts could be clustered together. But generally we opt for nuclear genes analysis to fully resolve evolutionary position of Pleurozia.

Methods

(11, 412-413) Based on the authors’ reply to the previous query, it seems that, for the phylogeny inference, they used sequences encoded by the mitogenomes, i.e., in their pre-editing state. This is a questionable choice, as the sequences prior to editing are non-functional. Using the (predicted) edited transcripts sequences or protein sequences (obtained from edited transcript sequences) would have been more reliable. At the very least, a comparison of the results between the pre-edited and edited sequences would be appropriate.

The reason of such approach was not to observe the influence of RNA editing on mitogenomic-based phylogeny of liverworts and mosses but rather to investigate evolutionary pattern of RNA editing sites distribution.

(11, 413-414) If the order of steps—concatenation, then sequence alignment—was performed as stated by the authors, the results are uncertain. Concatenation followed by alignment is prone to artifacts. First, align each gene (or protein), then concatenate. In addition, prior to the phylogeny inference, one should remove dubiously aligned positions (e.g., using trimAl, G-blocks or similar tools).

The order of steps was: alignment of each gene then concatenation of alignments. It was not clearly stated in the manuscript - corrected.

(11, 415) Please specify which model was chosen as the best fit and what was the best partitioning scheme.

The best-fit models of evolution for each subset as well as the best partitioning scheme are now provided in the Table S9.

(11, 418) The burn-in is better expressed as a percentage of the produced trees.

Corrected.

(11, 420) How were the ML analyses performed? Which algorithm and parameters? (In addition, Geneious does not provide a native ML tree-generation tool, but plugins to run RAxML, PhyML, or Fast-Tree.)

Corrected. The details are provided within manuscript Methods (11, 420-421).

(11, 421) Figure S1 — Even if the topology is the same, it would be nice if the tree was provided with the same species arrangement, so that the reader does not need to puzzle too much over it to see that the branching is congruent.

Corrected.

(12, 435) The argument provided in the response to the reviewer that ‘the script that was used in the study is still in developmental stage’ is unacceptable. This is like arguing that one will not provide a protocol for a wet-lab experiment, but wants to publish the results based on the said protocol. The scripts should be—at the very least—made available in the supplementary information, together with the relevant descriptions (please read the journal’s policies).

The Python script has been uploaded to GitHub (https://github.com/gymnomitrion/binary_editing.git) (12, 435-436).

Figure 1

1) For better readability, compress the ‘editing-site’ tree (on the right side) to ~2/3 and make the species names and editing site numbers (in the central part of the figure) larger.

2 ) The color scheme has not been changed at all (and thus remained colorblind unfriendly), despite authors' claim of having corrected that issue.

The ‘editing site’ tree has been compressed and the branch of the Bayesian inference tree has been shortened (5x). Also the color scheme has been changed to colorblind friendly.

Figure 2 & 3

1) If they the bar plots are arranged in the order of decreasing significance, state this in the legend. (It would be helpful.)

2) Deviations going into negative values do not make much sense. (How can the deviation be larger than the number of editing sites?)

Genes are sorted in ascending order of statistical significance of differences between editing frequencies. It is stated in the legend of Fig. 2 and Fig. 3. The bar plot whiskers now depict standard error values.

Typos and suggested text edits

(1, 19) “spann through almost” — should rather be “span almost”

(1, 29-30) “but also few genes are known of obligatory RNA editing” — ‘several’ instead of ‘few’ is more appropriate here; replace ‘known of’ with ‘known to undergo’

(2, 85) “including 18 liverworts and 18 mosses ” — ‘comprising’ would have been more appropriate (‘including’ makes it seem as if other species in addition to the 18+18 were analyzed)

(3, 97) “in the mitogenomes early land plants” — should be “in the mitogenomes of early land plants”

(5, 190) “The Fourteen” — remove ‘The’

(6, 214) “contrairly” — ‘contrarily’

(6, 220) “does not fit to the lost of editing sites” — ‘does not fit the loss of editing sites’

(7, 247) “were” — ‘where’

(11, 395) “n order” — ‘In order’

Corrected.

Round 3

Reviewer 2 Report

This version of the manuscript is far superior to the original and first revision. and the authors have adequately addressed the noted issues.  There are still a few grammatical errors (e.g. sentence fragment on lines 274-276) that can be corrected at the copy-edit stage.  

Author Response

The noted issues were addressed in the recent version of manuscript. Thank you for all  suggestions.

Reviewer 4 Report

In their revised manuscript, Myszczyński et al. made additional improvements by addressing most issues raised by the reviewers, but some questions requiring refinement have remained. Although most issues are now minor, one methodological aspect (detailed below) should first be addressed before the manuscript is accepted.

Methodology-related issue

The authors correctly point out the need to normalize the effects of RNA editing on genes in order to compare editing  frequencies across the numerous analyzed species. They chose to normalize according to gene length. There are, however, two issues.

1) It should be specified, in the Methods, whether the entire length of the gene (i.e., including introns) was used, or rather merely the CDS length. If the latter (which makes more sense), please correct the current ‘definition’.

2) More importantly, I am not convinced that gene length alone is the appropriate normalization factor. It would make more sense to normalize the editing levels to the number of editable residues (i.e., the number of Cs for C-to-U editing). One obvious advantage of such a normalization is that any biases introduced by fluctuating GC-content are taken care of.

[As a note aside, there is example from the shotgun proteomics field, which has faced a similar conundrum: to compare protein abundances, the number of peptide detections should be normalized to avoid inflation due to protein size alone (bigger the protein, higher the chance of producing a peptide that can be detected after protease cleavage). The normalization can be performed to the length of the protein, to the protein's molecular weight, or to the number of peptides theoretically detectable for a given protein (a parameter which depends on the number of cleavage sites for a protease, most commonly trypsin). The normalization to detectable peptides turned out to be more accurate than the other two methods.]

Introduction

(page 1, lines 30-31) “RNA editing not only increases genetic diversity and adaptation [5], but also, for some genes, it is necessary for their correct functioning [6]. ” — The phrasing seems to indicate that RNA editing is primarily adaptative and diversifying, but only rarely rectifying potentially deleterious mutations. Unless someone quantifies the instances of the process across eukaryotes (which has not been done, to my knowledge), this is not appropriate.

(1, 36) REF. #12 is inappropriate; it does not mention intron editing (just editing in RNA and intron trans-splicing).

(1, 36-37) “RNA editing also occurs at sites where the editing substitution is not relevant for the encoding protein function [13]” — Such a simplified interpretation is not without problems (which the authors of the original paper acknowledge), as the editing event could be required under certain conditions (developmental, environmental, etc.).

(1, 39) & (2, 66) REF. #16 is inappropriate. Please provide a reference to a study that actually made transcriptome-based validation. The one that the authors refer to, made only predictions and thus cannot be considered relevant in this context.

(2, 92-93) “The mitogenome-wide approach provided insight into the conservativeness of editing site distribution among genes and species.” — As mentioned previously, it is better to avoid vague phrasing, which does not specify, what was actually observed; rather provide statements with some substance (e.g., “we observed that editing site distribution is/is not conserved among genes and among species”, etc.).

(3, 96-97) “Is the reverse RNA editing (U-to-C) evolutionary pattern similar that typically found in the mitogenomes of early land plants?” — Inappropriate; this study does not provide any relevant answer to this question. (Plus, as the authors admit themselves, the U-to-C predictions are hardly well grounded.)

Results & Discussion

(4, 141) “thalloid liverworts, Blasia pusilla and Riccia fluitans, revealed the lowest editing potential” — Please reformulate; the authors through their analyses, not the liverworts themselves, revealed what the editing potential is.

(5, 165-169) Regarding the role of retro-processing in editing site loss — The authors mention in the text that 1) leafy liverwort genes contain introns, 2) they could be losing their introns by retro-processing, and 3) this could be connected to the decrease in the (predicted) number of editing sites. However, it is unclear on what kind of data the rationale is based. For any reader to follow the logic behind the authors’ reasoning, crucial information is missing: What is the intron number in simple thalloids? What is the position of these introns? Are the lost introns (in leafy liverworts) at all at positions where one can seriously consider that retro-processing could be involved? My previous question in this topic was not satisfactorily addressed in the authors’ reply. The discussion should put forward explanations why the analysis is currently impossible: what are the limitations and how can they be addressed (beyond reliance on pure luck, which the authors suggest in their response)?

(6, 195-196) “Genes are sorted in ascending order of statistical significance of differences between editing frequencies.” — My fault for not clearly asking this previously: ascending from left to right or right to left?

(8, 295-296) The authors allude to a moderately positive correlation of editing frequencies between liverworts and mosses. It would be useful to plot the frequencies to visualize the correlation and show outliers. Currently, it is not clear why the authors chose to single out all nad genes. The visual inspection of Figure 4 (to which the authors point in the following text) indicates that the entire deviation from the observed editing frequency similarity could actually be hijacked by merely three outlier genes: nad3, nad4L, and nad6. (By the way, it is suspicious that these are the shortest nad genes.) In addition, based on the Figure 4 as is, by eye, rpl (and to some extent rps) genes in mosses seem affected more by editing than their liverwort counterparts. Obviously, in order to assess the relevance of the plot, statistical validation is necessary. What is the Pearson correlation for rpl genes? Is it significantly different from nad genes?

(9, 321) “nad family members, seem to present a taxon-dependent RNA editing pattern” — Another vague phrasing; please explain what the claimed pattern is.

(9, 327-329) “The results obtained in the current study show that atp9 gene displays a high C-to-U RNA editing frequency in bryophytes” — This is not appropriate phrasing. Unless wet-lab experiments are provided, claims such as this have to be toned down.

(10, 353-359) Regarding the results of a phylogenetic analysis based on editing site numbers — There are numerous problems with this section. 1) The current analysis merely shows that the number of predicted editing sites in Pleurozia is more similar to those of other leafy liverworts. Presenting it as if its phylogenetic position was for the first time resolved and with good support is misleading. 2) The results of this analysis show a direct consequence of such an evolutionary process having played out in the first place (under the assumption that the editing site predictions are valid). Phrasing it as “in this case may reveal an interesting process” is inappropriate. 3) There seems to be some confusion regarding the impacts of RNA editing when the authors claim that “the RNA-editing process could reduce distances of Pleurozia and remaining leafy liverworts”. The process of RNA editing rectifies a divergent sequence (at least in cases, which are considered here, in plant organelles), so it reduces phylogenetic distances. Yet, there is less RNA editing in Pleurozia (at least based on the predictions), so what is reduced is the reliance on the RNA editing process. It is actually the evolutionary process opposite to the RNA editing that reduces the distances in the kind of analysis that the authors performed.

Methods

(12, 434) Without any identification which number corresponds to which gene, these the two new supplementary tables are practically useless.

Conclusions

(12, 470-472) Phrasing it as “U-to-C editing prediction may require improvement” is quite an understatement. The validation rates showed that the predictions are far from perfect for C-to-U (which should be acknowledged here, too), but clearly unreliable for U-to-C editing.

Figure 1

Currently chosen blue shades are too dark, making the tree difficult to read. (The color choice seems to cause unnecessary issues; if the authors are open to suggestion, they could use the following three (HEX code): CAE9F8, A5D9F3, 81CAEE.)

Typos and suggests text edits

(12, 461) revealed that C-to-U RNA editing — should rather be “revealed that the level of C-to-U RNA editing”

(12, 466) seems to be consistent for each gene — should rather use “similar” instead of “consistent”

Author Response

In their revised manuscript, Myszczyński et al. made additional improvements by addressing most issues raised by the Reviewers, but some questions requiring refinement have remained. Although most issues are now minor, one methodological aspect (detailed below) should first be addressed before the manuscript is accepted.

Methodology-related issue

The authors correctly point out the need to normalize the effects of RNA editing on genes in order to compare editing  frequencies across the numerous analyzed species. They chose to normalize according to gene length. There are, however, two issues.

1) It should be specified, in the Methods, whether the entire length of the gene (i.e., including introns) was used, or rather merely the CDS length. If the latter (which makes more sense), please correct the current ‘definition’.

Corrected.

2) More importantly, I am not convinced that gene length alone is the appropriate normalization factor. It would make more sense to normalize the editing levels to the number of editable residues (i.e., the number of Cs for C-to-U editing). One obvious advantage of such a normalization is that any biases introduced by fluctuating GC-content are taken care of.

[As a note aside, there is example from the shotgun proteomics field, which has faced a similar conundrum: to compare protein abundances, the number of peptide detections should be normalized to avoid inflation due to protein size alone (bigger the protein, higher the chance of producing a peptide that can be detected after protease cleavage). The normalization can be performed to the length of the protein, to the protein's molecular weight, or to the number of peptides theoretically detectable for a given protein (a parameter which depends on the number of cleavage sites for a protease, most commonly trypsin). The normalization to detectable peptides turned out to be more accurate than the other two methods.]

The previous studies on presence of editing sites in plant mitogenomes did not use normalization at all or used length as normalization factor (Mower & Palmer 2006 (doi.org/10.1007/s00438-006-0139-3), Sloan et al. 2010 (doi.org/10.1534/genetics.110.118000), Rudinger et al. 2011 (doi.org/10.1007/s00239-012-9486-3), Edera et al. 2018 (doi.org/10.1007/s11103-018-0734-9), Brenner et al. 2019 (doi.org/10.1534/g3.118.200763)). We admit, that we were thinking about normalizing not on length but on the cytosine numbers, as suggested by Reviewer. But due to the fact that editing site recognition is not based on the simply C presence but require 20-25 upstream nucleotides before edited cytosine (Shikanai 2015 doi.org/10.1016/j.bbabio.2014.12.010) we have chosen CDS length as a normalization factor. For instance if we have hypothetical 102 bp genes, there could be 4 to 5 editing sites, so there can be up to five cytosines edited, even if there was only 5 in the whole gene, but is unlikely the all of them will be edited if they appeared as homopolymer. But the cytosine frequence will be equal in both cases. The second reason was the lack of correlation between mean GC content of genes and mean number of editing sites within genes (r = 0.124, Pearson correlation, p-value = 0.4918): the highest GC content was found in the rps12 gene (mean GC content = 43%) which was one of the least edited genes, while tatC (mean GC content = 33.4%) was the one of most edited.

Introduction

(page 1, lines 30-31) “RNA editing not only increases genetic diversity and adaptation [5], but also, for some genes, it is necessary for their correct functioning [6]. ” — The phrasing seems to indicate that RNA editing is primarily adaptative and diversifying, but only rarely rectifying potentially deleterious mutations. Unless someone quantifies the instances of the process across eukaryotes (which has not been done, to my knowledge), this is not appropriate.

Rephrased. “The modifications of transcripts caused by the RNA editing effect with encoding of alternative amino acid sequences which is necessary for correct functioning of some protein-coding genes [5]. The RNA editing may be also involved in increasing genetic diversity and adaptation [6].

(1, 36) REF. #12 is inappropriate; it does not mention intron editing (just editing in RNA and intron trans-splicing).

Reference replaced with another one, which states: :”To our knowledge, this is the first experimental evidence that RNA editing of a C residue is mandatory for the splicing of a mitochondrial mRNA precursor in conditions closer to the in vivo situation” (Castandet et al. 2010).

(1, 36-37) “RNA editing also occurs at sites where the editing substitution is not relevant for the encoding protein function [13]” — Such a simplified interpretation is not without problems (which the authors of the original paper acknowledge), as the editing event could be required under certain conditions (developmental, environmental, etc.).

Rephrased: “On the other hand, RNA editing also occurs at sites where the editing substitution is not essential for protein expression or activity. However the occurrence of RNA editing substitutions might vary across living conditions i.e. stage of development, environmental conditions [13]”.

(1, 39) & (2, 66) REF. #16 is inappropriate. Please provide a reference to a study that actually made transcriptome-based validation. The one that the authors refer to, made only predictions and thus cannot be considered relevant in this context.

Corrected. Since the higher abundance of RNA editing in plastomes of liverworts has not been confirmed with the transcriptome-based study the sentence was removed.

(2, 92-93) “The mitogenome-wide approach provided insight into the conservativeness of editing site distribution among genes and species.” — As mentioned previously, it is better to avoid vague phrasing, which does not specify, what was actually observed; rather provide statements with some substance (e.g., “we observed that editing site distribution is/is not conserved among genes and among species”, etc.).

Sentence removed.

(3, 96-97) “Is the reverse RNA editing (U-to-C) evolutionary pattern similar that typically found in the mitogenomes of early land plants?” — Inappropriate; this study does not provide any relevant answer to this question. (Plus, as the authors admit themselves, the U-to-C predictions are hardly well grounded.)

Sentence removed.

Results & Discussion

(4, 141) “thalloid liverworts, Blasia pusilla and Riccia fluitans, revealed the lowest editing potential” — Please reformulate; the authors through their analyses, not the liverworts themselves, revealed what the editing potential is.

Rephrased. "Besides the known, non-editing model liverwort genus - Marchantia [31], two other complex thalloid liverworts, Blasia pusilla and Riccia fluitans, were found to have the lowest editing potential among liverworts."

(5, 165-169) Regarding the role of retro-processing in editing site loss — The authors mention in the text that 1) leafy liverwort genes contain introns, 2) they could be losing their introns by retro-processing, and 3) this could be connected to the decrease in the (predicted) number of editing sites. However, it is unclear on what kind of data the rationale is based.

Ad 1) Yes, leafy and thallose (simply and complex) contains introns and in the most of the species the intron content is stable. There is no fluctuation of intron content in thalloid liverworts.

Ad 2) In some leafy liverworts certain introns are missing, and the missing one are precisely cut, not reduced, with loss of the adjacent editing sites (Ślipiko et al. 2017), this suggest that retro-processing is involved (Cuenca et al. 2016)).

Ad 3) It is difficult to point out retro-processing as a mechanism of editing sites loss if we have no proof of it (retro-processing) in the most of the genes. It is partially possible in intron-containing genes with detected intron loss that fit into retro-processing scenario.

For any reader to follow the logic behind the authors’ reasoning, crucial information is missing: What is the intron number in simple thalloids? What is the position of these introns? Are the lost introns (in leafy liverworts) at all at positions where one can seriously consider that retro-processing could be involved? My previous question in this topic was not satisfactorily addressed in the authors’ reply. The discussion should put forward explanations why the analysis is currently impossible: what are the limitations and how can they be addressed (beyond reliance on pure luck, which the authors suggest in their response)?

These data were published in previous studies concerning general structures of bryophytes mitogenomes (Liu et al. 2012, Liu et al. 2014, Myszczyński et al. 2017, Ślipiko et al. 2017, Myszczyński et al., 2018, Dong et al. 2019) and the analysis of changes in the intronic content was not the main target of this manuscript. It very interesting process, however it is out of the scope of this study. Furthermore, it certainly would require much wider sampling of the leafy liverworts.

(6, 195-196) “Genes are sorted in ascending order of statistical significance of differences between editing frequencies.” — My fault for not clearly asking this previously: ascending from left to right or right to left?

The borderline of p-value suggests the order of sorting, however we have additionally provided the information in the Fig. 2 and Fig. 3 caption (left to right).

(8, 295-296) The authors allude to a moderately positive correlation of editing frequencies between liverworts and mosses. It would be useful to plot the frequencies to visualize the correlation and show outliers. Currently, it is not clear why the authors chose to single out all nad genes. The visual inspection of Figure 4 (to which the authors point in the following text) indicates that the entire deviation from the observed editing frequency similarity could actually be hijacked by merely three outlier genes: nad3, nad4L, and nad6. (By the way, it is suspicious that these are the shortest nad genes.) In addition, based on the Figure 4 as is, by eye, rpl (and to some extent rps) genes in mosses seem affected more by editing than their liverwort counterparts. Obviously, in order to assess the relevance of the plot, statistical validation is necessary. What is the Pearson correlation for rpl genes? Is it significantly different from nad genes?

The editing frequencies regarding majority of studied liverworts and mosses species are presented on previous figures (Fig. 2 and Fig. 3). We prefer to avoid presenting almost identical content. According to the Reviewer suggestion we have resigned from distinguishing whole nad gene family as the reason of editing frequency similarity deviation. Instead, we emphasized the observation that the correlation could be distorted by single genes such as: nad2, nad4L, nad6, rps7 or rps14. The Pearson correlation of the rest of the genes was 0.866 (p-value = 2.543e-09). The Pearson correlation for rpl genes (regarding Reviewer request), was 0.990 (p-value = 0.0011). The correlation for the all nad genes was 0.147, however the p-value = 0.7273 depicted that the correlation was insignificant (there was no correlation). We also showed the significance value while providing Pearson correlation value of complete set of 33 genes.

(9, 321) “nad family members, seem to present a taxon-dependent RNA editing pattern” — Another vague phrasing; please explain what the claimed pattern is.

Sentence rephrased: “Although the majority of mitochondrial genes investigated in this study seem to be in accordance with this hypothesis, for some genes, like nad2, nad4L, nad6, rps7 and rps14, RNA editing seems to be taxon-dependent”.

(9, 327-329) “The results obtained in the current study show that atp9 gene displays a high C-to-U RNA editing frequency in bryophytes” — This is not appropriate phrasing. Unless wet-lab experiments are provided, claims such as this have to be toned down.

Corrected: "The results obtained in the current study show that atp9 gene displays a high potential of C-to-U RNA editing frequency in bryophytes".

(10, 353-359) Regarding the results of a phylogenetic analysis based on editing site numbers — There are numerous problems with this section. 1) The current analysis merely shows that the number of predicted editing sites in Pleurozia is more similar to those of other leafy liverworts. Presenting it as if its phylogenetic position was for the first time resolved and with good support is misleading. 2) The results of this analysis show a direct consequence of such an evolutionary process having played out in the first place (under the assumption that the editing site predictions are valid). Phrasing it as “in this case may reveal an interesting process” is inappropriate. 3) There seems to be some confusion regarding the impacts of RNA editing when the authors claim that “the RNA-editing process could reduce distances of Pleurozia and remaining leafy liverworts”. The process of RNA editing rectifies a divergent sequence (at least in cases, which are considered here, in plant organelles), so it reduces phylogenetic distances. Yet, there is less RNA editing in Pleurozia (at least based on the predictions), so what is reduced is the reliance on the RNA editing process. It is actually the evolutionary process opposite to the RNA editing that reduces the distances in the kind of analysis that the authors performed.

Ad 1) No, the number of editing site was not the subject of phylogenetics analysis, as we pointed it out in the Methods section, so in this case it did not influence the phylogeny - the P. platyphylla and G. concinnatum have similar numbers of editing sites but they were not clustered together. In the opposite case - A. pinguis (399 predicted editing sites) and M. furcata (233 predicted editing sites) were resolved as a sister species despite almost two fold difference in the numbers of predicted editing sites. In the corrected version we clearly stated that phylogeny is based on the RNA editing site distribution pattern, not on their numbers. It was described in Methods section, but since it is placed at the end of manuscript, we agree with the Reviewer, that it could be somehow misleading.

Ad 2 & 3) We agree with the Reviewer at this points and made corrections in the manuscript.

Methods

(12, 434) Without any identification which number corresponds to which gene, these the two new supplementary tables are practically useless.

Corrected. The gene names are provided within mentioned supplementary tables.

Conclusions

(12, 470-472) Phrasing it as “U-to-C editing prediction may require improvement” is quite an understatement. The validation rates showed that the predictions are far from perfect for C-to-U (which should be acknowledged here, too), but clearly unreliable for U-to-C editing.

Throughout the manuscript we have clearly stated that the reader should treat U-to-C predictions with caution. To ultimately confirm unreliability of U-to-C predictions the predictions should be confronted with experimentally confirmed corresponding transcriptomes which is not the case.

Sentence changed to: “U-to-C editing prediction require further improvement”.

Figure 1

Currently chosen blue shades are too dark, making the tree difficult to read. (The color choice seems to cause unnecessary issues; if the authors are open to suggestion, they could use the following three (HEX code): CAE9F8, A5D9F3, 81CAEE.)

Corrected.

Typos and suggests text edits

(12, 461) revealed that C-to-U RNA editing — should rather be “revealed that the level of C-to-U RNA editing”

Corrected.

(12, 466) seems to be consistent for each gene — should rather use “similar” instead of “consistent”

Corrected.